## REPORT

# Actin arginylation alters myosin engagement and F-actin patterning despite structural conservation

Clyde Savio Pinto[1]*, Saskia E. Bakker[2]*, Andrejus Suchenko[1]*, Isabella M. Kolodny[3], Hamdi Hussain[1]**, Tomoyuki Hatano[1]**, Karuna Sampath[1], Krishna Chinthalapudi[3], Sarah M. Heissler[3], Masanori Mishima[1], and Mohan Balasubramanian[1]

Actin is a conserved protein with crucial roles in cell polarity, division, and muscle contraction. Its function is regulated in part by posttranslational modifications, one of which is N-terminal arginylation. What is the structure of arginylated-β-actin (R-β-actin), and how does it regulate F-actin function? Here we report the 3.6 Å structures of ADP-R-β-actin filaments, which are nearly identical to that of non-arginylated F-actin. *In vitro* assays reveal that the interaction between myosin-II and actin is altered upon actin arginylation, characterized by frequent detachment of R-actin filaments from myosin-II. *In vivo*, replacement of the only actin gene in *Schizosaccharomyces pombe* with a synthetic gene encoding R-Sp-actin reduces Arp2/3-based actin patches while thickening formin-induced actin cables. Consistent with defective interactions between myosin-II and R-actin filaments, assembly and constriction of the cytokinetic actomyosin ring are perturbed in R-Sp-actin cells. Thus, despite structural similarity of arginylated and non-arginylated actin filaments, actin arginylation affects F-actin assortment into distinct subcellular structures and its interaction with myosin-II.

## Introduction

Actin is one of the most conserved and abundant eukaryotic cytoskeletal proteins and plays a key role in cell motility, division, polarity, and muscle contraction (Dominguez and Holmes, 2011; Holmes, 1998; Kabsch and Holmes, 1995; Pollard and Cooper, 2009). The structure of actin filaments has been determined by cryogenic electron microscopy (cryo-EM), revealing near-atomic resolution actin filament structures (Arora et al., 2023; Avery et al., 2017; Belyy et al., 2020; Chou and Pollard, 2019; Kumari et al., 2020; Raunser, 2017; von der Ecken et al., 2016), which have built on beautiful work elucidating the structure of actin monomers using X-ray crystallography (Holmes et al., 1990; Schutt et al., 1993). Actin organizes into two helical strands, which undergo continuous polymerization and depolymerization. Actin isoforms have posttranslational modifications that appear to facilitate specific cellular functions (Terman and Kashina, 2013; Varland et al., 2019). Among these modifications for the human non-muscle (NM) β-actin are the N-terminal acetylation and His-73 methylation (Terman and Kashina, 2013; Varland et al., 2019). In addition, a subset of β-actin undergoes a process termed arginylation in which an arginine residue is added to the N terminus (Karakozova et al., 2006; Kashina, 2014; Lian et al., 2014; Terman and Kashina, 2013). It is now clear that His-73 methylation, carried out by

the SETD3 family of methyltransferases, regulates Pi release following ATP hydrolysis within the filament (Dai et al., 2019; Hintzen et al., 2021; Kwiatkowski et al., 2018; Lappalainen, 2019; Wilkinson et al., 2019). N-terminal acetylation of actin, mediated by the NAA80 enzyme, plays important roles in actin polymerization and depolymerization kinetics and affects cellular structures such as filopodia and lamellipodia (Aksnes et al., 2018; Arnesen et al., 2018; Beigl et al., 2020; Drazic et al., 2018; Drazic et al., 2022; Ree et al., 2020). β-actin arginylation, which is achieved by the arginyl-tRNA-protein transferase Ate1 following the removal of the N-terminal acetylated aspartic acid residue, has been proposed to regulate lamellipodia formation, cell size, and cell spreading, but the exact biochemical and structural mechanisms are unknown (Karakozova et al., 2006; Saha et al., 2010).

In this study, to address the structural and physiological mechanisms impacted by actin arginylation, we used single-particle cryo-EM combined with *in vitro* analysis of the actin–myosin interaction as well as *in vivo* analysis in the genetically tractable fission yeast *Schizosaccharomyces pombe.* Our findings identify differences between arginylated and non-arginylated forms in actin filament organization and interactions of the filaments with myosin.

[1]Centre for Mechanochemical Cell Biology and Division of Biomedical Sciences, Warwick Medical School, Coventry, UK;   [2]School of Life Science, University of Warwick, Coventry, UK;   [3]Department of Physiology and Cell Biology, Dorothy M. Davis Heart and Lung Research Institute, The Ohio State University College of Medicine, Columbus, OH, USA.

*C.S. Pinto, S.E. Bakker, and A. Suchenko contributed equally to this paper;   **H. Hussain and T. Hatano contributed equally to this paper.   Correspondence to Masanori Mishima: m.mishima@warwick.ac.uk;   Mohan Balasubramanian: m.k.balasubramanian@warwick.ac.uk.

## Results and discussion

### No drastic effect of arginylation on actin and actin filament structures

Using a method that we recently developed, we purified from *Pichia pastoris* large amounts of recombinant human arginylated-β-actin (R-β-actin) (Hatano et al., 2018) in which His73 was methylated by co-expression of human SETD3 (Hatano et al., 2020). We assessed if this R-β-actin assembled into filaments with equal efficiency compared with Ac-β-actin, as determined using a pelleting assay. No significant difference was detected between R-β-actin and Ac-β-actin after polymerization for 1 h in the fraction of filaments sedimented following centrifugation (Fig. S1 A). To assess the effect of N-terminal arginylation on the actin filament, we solved the structure of fully arginylated actin filaments by cryo-EM. Globular R-β-actin was polymerized, plunge-frozen, and screened to evaluate filament formation and ice thickness. Following optimization of filament assembly conditions, we collected 2,285 images where the actin filaments are clearly identifiable in the raw micrographs (Fig. 1 A). Individual filament images were boxed into overlapping segments, and 50 classes were generated. The wide and narrow projections of the filaments are clearly resolved in the class averages (Fig. 1 B). Following this, using the helical reconstruction approach (He and Scheres, 2017), we obtained a 3D map of the ADP-bound human R-β-actin to a resolution of 3.6 Å for the subunit (Fig. 1 C). In this map, displayed with the pointed end of the filament positioned to the top, we observed well-resolved secondary structure elements. Local resolutions range from 2.2 Å for the core region to 3.6 Å for the flexible, solvent-exposed areas, including the DNase I–binding loop (D-loop) and N terminus (Fig. S1 B). The refined map shows a helical twist of 166.8° and a rise of 28.1 Å, consistent with values reported previously (Fig. 1 C) (Arora et al., 2023; Chou and Pollard, 2019; Galkin et al., 2015).

An atomic model was built into the density map using a rigid-body fit of the published chicken skeletal muscle actin (Galkin et al., 2015) as a starting point. At the present resolution, we were able to reliably trace the backbone (Fig. 1 C) and resolve the density of the side chains of the larger residues (Fig. 1 D) and $Mg^{2+}$- ADP (Fig. 1 E). Compared with the core of the subunit, we observe a weaker backbone density for residues Met46 and Gly47 in the D-loop (Fig. 1 D). The 2 N-terminal residues are not resolved in the electron density (Fig. 1 F), indicating the N terminus is flexible. This is consistent with the lower resolution of this region, as shown by the local resolution analysis.

The structure of the arginylated human F-actin is consistent with previously published F-actin structures, including the recently published human β-actin structure (Arora et al., 2023) (Fig. 2). The root-mean-square deviation (RMSD) between R-β-actin and human Ac-β-actin (8DNH) is 0.851 Å (Fig. 2 A), and the active site is consistent with previous structures (Fig. 2 B). As expected, higher RMSD is seen in the D-loop region around residue 50, and two solvent-exposed loops around residues 146 and 245, corresponding to the areas with lower resolution as shown in the local resolution map (Fig. S1 B). A comparison of the N-terminal residues shows the three negatively charged Asp residues in β-actin, and these negative charges are reduced by the removal of an aspartate and introduction of a positively charged arginine in R-actin (Fig. 2 C). We then compared R-β-actin with previously published chicken and rabbit skeletal muscle F-actin (Fig. S1, C–E) (Chou and Pollard, 2019; Galkin et al., 2015). We used CCP4 Superpose by Secondary Structure Matching (Krissinel and Henrick, 2004) to establish matching residue ranges. Residues 3-374 in the structure presented here were matched with residues 4–375 in structures 6DJO and 3J8I. This shows a backbone RMSD of 0.965 Å between the R-β-actin filament structure determined in this work and 6DJO (chicken, [Chou and Pollard, 2019]) and 1.248 Å between the structure of R-β-actin and 3J8I (rabbit, [Galkin et al., 2015]). Despite the higher sequence similarity between the rabbit and chicken actin models, the RMSD between the two at 1.353 Å is higher than between either of those structures and our model of human F-actin.

### Arginylation impairs the interaction of F-actin with myosin-II

Arginylation of β-actin at the N terminus exposed by removal of the acetylated aspartate increases the electrostatic surface charge by three as compared with Ac-β-actin. In the actomyosin complex, the N terminus of actin is closely located to the positively charged surface patches on myosin (Fig. 3 A). The N-terminal residues of actin have been reported to be involved in the functional/physical interaction with myosin (Arora et al., 2023; Behrmann et al., 2012; von der Ecken et al., 2016). Thus, we then examined the influence of actin arginylation on the actin–myosin interaction using *in vitro* assays with purified proteins.

When loaded onto NM myosin-IIA–coated coverslips in the absence of ATP, the number of filaments attached and their length distributions were similar between the R-β-actin and Ac-β-actin filaments (Fig. S2, A–C). These observations are consistent with the results of the bulk sedimentation assay (Fig. S1 A) and saturation levels of polymerization reported previously (Chin et al., 2022), though the kinetics of polymerization was reduced by a factor of ~1.5–1.8. These actin preparations with comparable polymerization levels and filament length distributions were used in the following *in vitro* assays.

Interaction with actin stimulates the ATPase activity of the myosin motor domain, which releases energy for mechanical work. We assessed the actin-activated ATPase activity of NM myosin-II by measuring the increase of free phosphate using a spectrophotometric assay. Although both polymerized and phalloidin-stabilized Ac-β-actin and R-β-actin promoted myosin ATPase activity in concentration-dependent manners, the degree of stimulation by R-β-actin was about 40% that of Ac-β-actin (Fig. 3 B). This indicates that R-β-actin is less effective in activating the ATPase activity of NM myosin-II compared with Ac-β-actin.

Myosin is an actin-based motor protein that translocates actin filaments in an ATP-dependent manner. We investigated the influence of the arginylation on myosin motor activity by observing the motility of R-β-actin and Ac-β-actin filaments on the coverslip surface coated with NM myosin-II (Fig. 3, C–H, and Video 1). While slightly more R-β-actin filaments were observed per field of view (Fig. 3 C), we observed a lower percentage of motile R-actin filaments compared with Ac-β-actin filaments

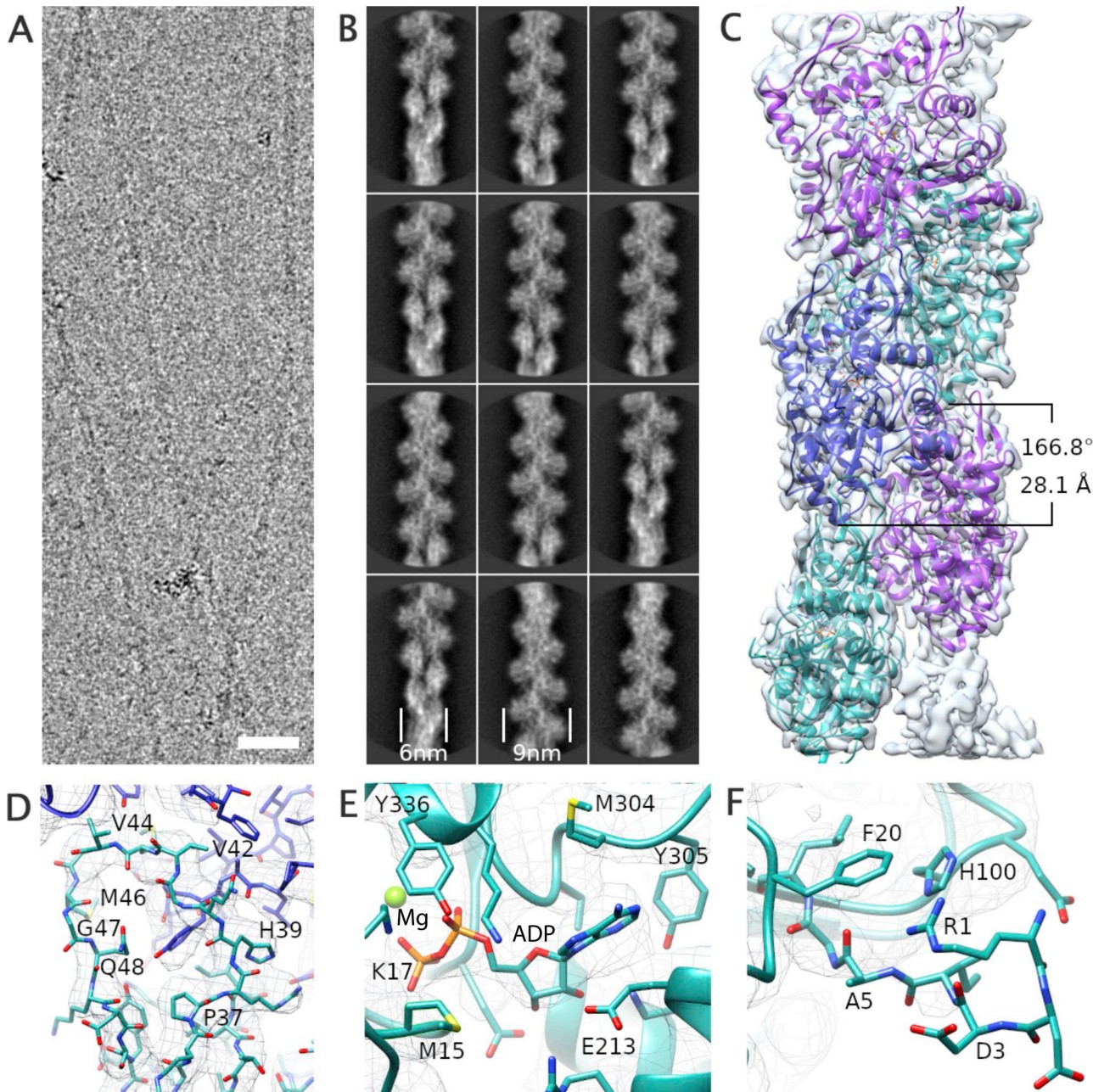

Figure 1.   **The structure of R-β-actin. (A)** Representative cryo-TEM image of human filamentous R-β-actin. Scale bar 20 nm. **(B)** The 2D class averages were obtained from 50 individual regions selected from the filaments, showing the wide and narrow projections of filaments (indicated with lines). **(C)** 3D reconstruction of human R-β-actin showing five actin subunits in different colors. **(D)** The D-loop of R-β-actin shows the fit of the protein backbone in the density. Residues in this region are shown in stick representation. **(E)** Ribbon diagram of the filamentous R-β-actin ADP-binding site, with ADP and interacting residues shown in stick representation. **(F)** The N terminus region of R-β-actin from residue Ile-4 is shown. Electron density fit to the residues 1 to 3 is missing, as this region is not resolved.

(Fig. 3 D). Nonetheless, the velocities of the motility were comparable (Fig. 3 E). Interestingly, the R-β-actin filaments showed abrupt displacement ("wobbling") or detachment from the surface much more frequently than the Ac-β-actin filaments on NM myosin II (Fig. 3, F–H and Fig. S2 D) as well as tissue-isolated cardiac myosin–coated surfaces (Fig. S2, D and E). These indicate that arginylation makes the interaction of the actin filament with myosin-II less efficient. Collectively, these observations suggest that the negative charge in the N-terminal actin tail,

which is reduced by the arginylation, contributes to persistent and productive interaction between F-actin and myosin-II, preventing unfavorable detachment of actin filaments from myosin-II.

**Cells that have R-Actin as the sole actin are viable but show an altered actin distribution and actomyosin ring function**
The structural and biochemical work strongly suggested that actin arginylation compromises actin–myosin-II interaction. We

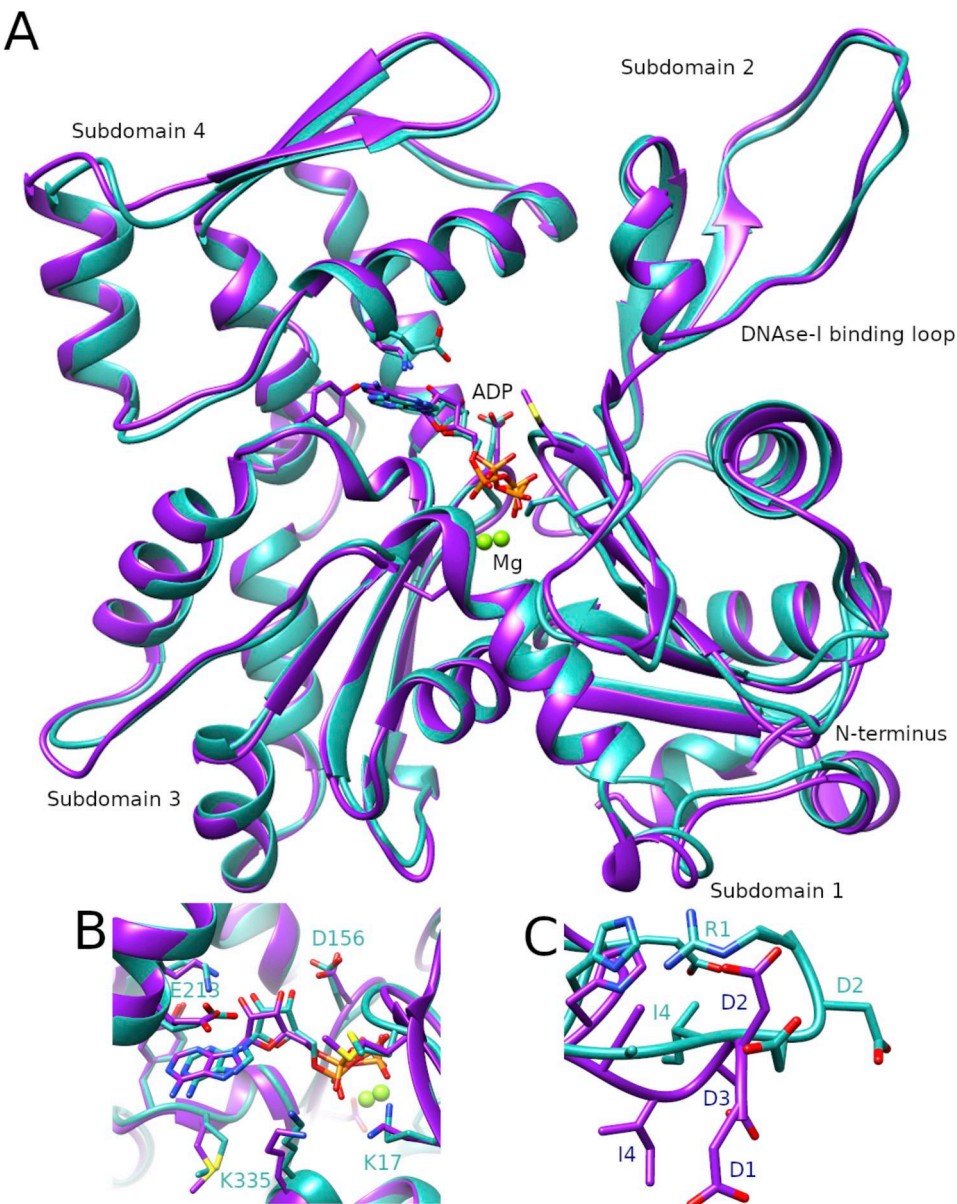

Figure 2. **Comparison of the R-β-actin and Ac-β-actin structures. (A)** Ribbon diagram comparing the structure of R-β-actin presented here (cyan) with the previously published structure of human Ac-β-actin (magenta). **(B)** Close-up view comparing the structure of the active site of R-β-actin with human filamentous Ac-β-actin. **(C)** Close-up view comparing the N-terminal sections of R-β-actin to that of human filamentous Ac-β-actin.

designed an *in vivo* experimental setup that allows us to assess the physiological consequences of replacing the entire cellular pool of actin with R-actin. Analysis of the function of actin modifications is complicated in mammalian cells due to the multiplicity of actin isoforms and modifications. To generate cells that expressed only R-actin, we used the fission yeast *S. pombe*. *S. pombe* has only one actin (*act1*), which does not undergo N-terminal maturation. It also has both Arp2/3 and formin-nucleated actin structures, making it an ideal system in which to study the effect of actin arginylation on the actin cytoskeletal organization and function. To express R-actin in fission yeast cells, we used our established strategy to express and purify R-actin from the methylotrophic yeast, *P. pastoris* (Hatano et al., 2018). Toward this goal, we engineered the endogenous *act1*

locus of *S. pombe* so that it expresses a fusion protein of an Ubi4 tag and either wild-type (wt) actin or R-actin. The Ubi4 tag is cleaved off inside the cell, resulting in either wt-Act1 or R-Act1, *S. pombe* actin with the N-terminal tail starting with MEEE and REE, respectively (Fig. S3). We modified the *act1* locus in the wt background and in strains expressing *rlc1*-3GFP and mCherry-*atb*2 (referred to as "*rlc1-GFP*" hereafter) as markers of the actomyosin ring and the mitotic spindle, respectively. The successful gene editing was confirmed by PCR and sequencing (Fig. S4, A and B).

Efficient removal of the Ubi4 tag (8.7 kDa) in *S. pombe* cells was confirmed by western blotting, which showed only a single band at the expected size for actin (42 kDa) (Fig. S4 C) in all the MEEEAct1 strains constructed. REEAct1 also appeared as a single

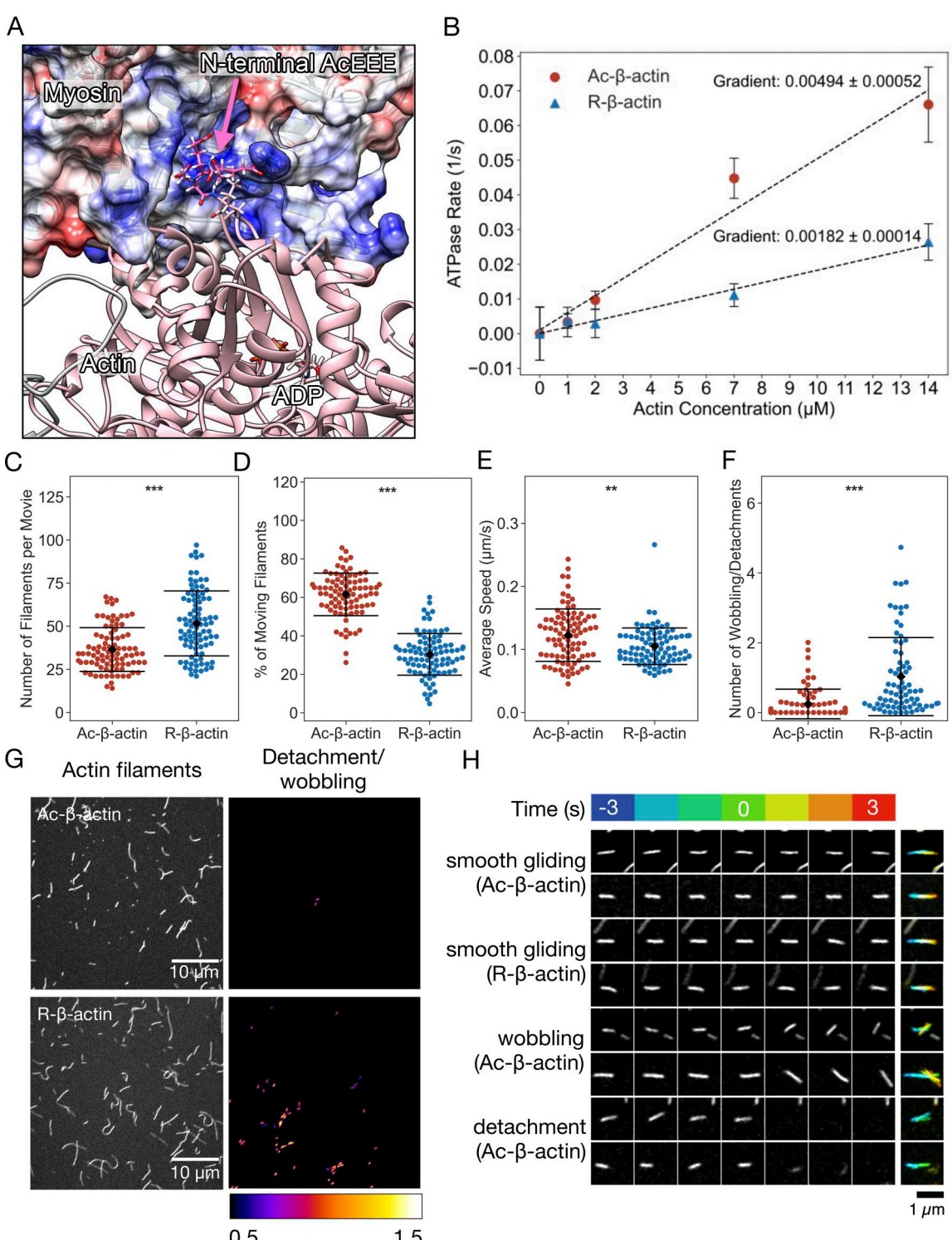

Figure 3. **The stability of the interaction between myosin and actin filaments is weakened by arginylation of actin. (A)** A model of the actomyosin complex, indicating the electrostatic interaction between the negatively charged actin N-terminal tail and the positively charged surface of myosin. A snapshot from a molecular dynamic simulation based on a cryo-EM structure (PDB:5JLH [von der Ecken et al., 2016]) is displayed with the N-terminal tail of gamma-actin (AcEEEI) in magenta and the surface coulombic potential of myosin IIc in a red (0 to –10 kcal/mol) and blue (0 to +10 kcal/mol) linear scale. **(B)** The steady-state rate of ATP hydrolysis by NM myosin measured in the presence of phalloidin stabilized Ac- or R-actin filaments. **(C–F)** The quantification of the myosin-driven motility of Ac- or R-actin filaments. Each dot corresponds to an experiment or a movie of 1 min duration. Mean (diamond) and the standard error (error bars) are indicated. **(C)** Number of filaments per movies. **(D)** Percentage of the moving filaments. **(E)** Average speeds of the moving filaments per movie. **(F)** Frequency of detachment or wobbling per filament. **(G)** The surface motility assay with immobilized NM myosin and Alexa Fluor-488–conjugated phalloidin-labelled filaments of Ac- or R-β-actin. The left panels show the total filaments landed in the presence of 1 mM ATP. The right panels indicate the detachment or wobbling of filaments. **(H)** Examples of the detachment and wobbling of actin filaments. Indicated time frames before and after the detected event were shown as well as the maximum projection with temporal color coding (blue to red).

band with a slightly higher mobility as shown previously with human R-β-actin purified from *P. pastoris* (Hatano et al., 2018). The N-terminal sequence of MEEEAct1 and REEAct1 in respective strains was also confirmed by mass spectroscopy (Fig. S4 D).

There was no obvious difference in the viability of *REEact1* and *MEEEact1* strains in spot assays (Fig. 4 A). However, the growth rate of the *REEact1* strains was slower than the *MEEEact1* strains with the *rlc1-GFP*–containing strains growing even slower (Fig. 4 B). To test if there were any size differences between the strains, we analyzed septated cells. The *REEact1* cells were shorter and wider with lower volumes and surface area (Fig. 4 C). To look at the actin cytoskeleton of these strains, we used phalloidin staining. Phalloidin staining in the wt background visually revealed a decrease in the area occupied by actin patches in the *REEact1* strain as opposed to the *MEEEact1* strain (Fig. 4 D). On the other hand, we qualitatively observed thicker cables in the *REEact1* strain (Fig. 4 D). To analyze patches in greater detail, we generated fimbrin-GFP strains that mark patches but not cables or rings. The patches were counted manually in 3D. This analysis showed that the number of patches in the *REEact1* strain is less than that in the *MEEEact1* strain per cell (Fig. 4 E (i)). Additionally, we also manually measured the area of well-separated patches in the slice in which they were brightest. This analysis showed that the area of patches is also decreased in the *REEact1* strain (Fig. 4 E (ii)).

Considering our *in vitro* findings that actomyosin interactions are perturbed, we studied cytokinesis, which is an actomyosin-driven process. Time-lapse observation revealed significant differences between the two strains in the timings of the cytokinesis stages (Fig. 5, A and B). The cytokinetic actin ring (CAR) in fission yeast is assembled from actomyosin nodes, which first appear and then coalesce into a ring (node coalescence phase), which then dwells without constricting for a time (dwell phase) and then constricts (ring constriction phase). Using myosin regulatory light chain fused to 3 GFPs (Rlc1-3GFP) as a marker for the CAR, we observed that, in the *REEact1* strain, nodes took longer to coalesce into a CAR, and the CAR dwelled for a shorter period than in the *MEEEact1* strain (Fig. 5, A and B; and Video 2). The ring constriction took longer in the *REEact1* strain, reflecting the reduced average rate of constriction (ring diameter divided by the time for constriction). Overall, the total time taken for cytokinesis from node appearance to the end of CAR constriction was increased in the *REEact1* strain. These observations, especially the slower ring constriction, suggest that myosin II function in the *REEact1* strain is altered, demonstrating a key role for arginylation in modulation of actin dynamics.

Given the frequent detachment and wobbling of R-β-actin filaments in gliding assays and the reduced CAR constriction rate in *REEact1* cells, we tested for genetic interactions. To this end, we crossed *MEEEact1* or *REEact1* with *myo2*-E1 mutants. *myo2*-E1 harbors a G345R substitution, compromised for interaction with wt actin, and is viable at 24°C but not at 36°C (Balasubramanian et al., 1998; Lord and Pollard, 2004; Palani et al., 2017). Tetrads were dissected (*n* = 13 in MEEE and 23 in REE), and meiotic products were allowed to resume vegetative growth at the permissive temperature for *myo2*-E1. We were unable to find any viable colony that contained both *REEact1* and

*myo2*-E1, although all other combinations were observed (Fig. 5 C). These results further strengthen the conclusion that arginylation of actin reduces its affinity for myosin-II and that the combinatorial effects of these weak viable alleles leads to synthetic lethality. We additionally tested for genetic interactions between *REEact1* and *myo1Δ*, *cdc12*-112, *for3Δ*, *wsp1Δ*, and *arp3*-C1. None of these combinations led to lethality (data not shown). Together, these observations suggest that R-Act1 can support viability as the sole actin in *S. pombe* but alters actin organization and impairs actomyosin ring assembly and constriction.

Collectively, in this study, we have resolved the structure of R-β-actin, discovered that arginylation of actin alters its interaction with NM myosin II, and developed a fission yeast system in which arginylated actin is the sole genetically provided actin. Our comparison of the R-β-actin structure with those from our previous work with Ac-β-actin as well as other previously published actin structures was not able to detect major differences in their overall structures. However, within our R-β-actin structure we were unable to resolve the N terminus. This suggests that the N termini of different subunits within the filament are in different conformations, indicating a higher flexibility in the N-terminal region. Despite the reduction in negative charge at the N terminus compared with acetylated actin, it appears the N-terminal arginylation does not help to stabilize the N terminus, at least in the ADP-bound structure. Our work also shows a less well-defined density in the D-loop, where adjacent subunits interact (Fig. 1 D) In previous studies, the D-loop was found to be more flexible in the ADP-bound structure of actin compared with structures containing AMPPNP or ADP-Pi for both chicken and rabbit actin (Chou and Pollard, 2019; Merino et al., 2018). This suggests the hydrolysis of ATP to ADP is a major contributor to this increased flexibility of the D-loop.

Previous work has shown functional differences between R-actin and acetylated actin in terms of polymerization and its functional interactions with actin-binding proteins such as the Arp2/3 complex (Chin et al., 2022). Since, the structure that we obtained for R-β-actin cannot account for these differences, though we were unable to sufficiently resolve the N terminus and the D-loop, we suggest that the charge difference on the N terminus in R-β-actin might be the cause of these differential interactions. Previous work has demonstrated that the quantum of negative charge on the N terminus is important for force generation through the interactions of the negatively charged actin N terminus with the positively charged myosin II (Arora et al., 2023; Behrmann et al., 2012; Lu et al., 2005; Miller et al., 1996; von der Ecken et al., 2016). Indeed, our *in vitro* experiments show that gliding R-β-actin filaments frequently detach from NM myosin-II, and actin-activated myosin ATPase activity is reduced. In general, myosin binds actin strongly when nucleotide-free or ADP-bound and weakly when bound to ATP or ADP–Pi. As a low-duty-ratio motor, myosin-II spends most of its ATPase cycle in the weak-binding state (Heissler and Sellers, 2016; Kovacs et al., 2003; Robert-Paganin et al., 2020). We speculate that the electrostatic interaction through the N-terminal tail of actin plays a role in loosely anchoring myosin heads in the weak or none-binding state, while its impact is minimal for the myosin in the strong-binding state (such as in

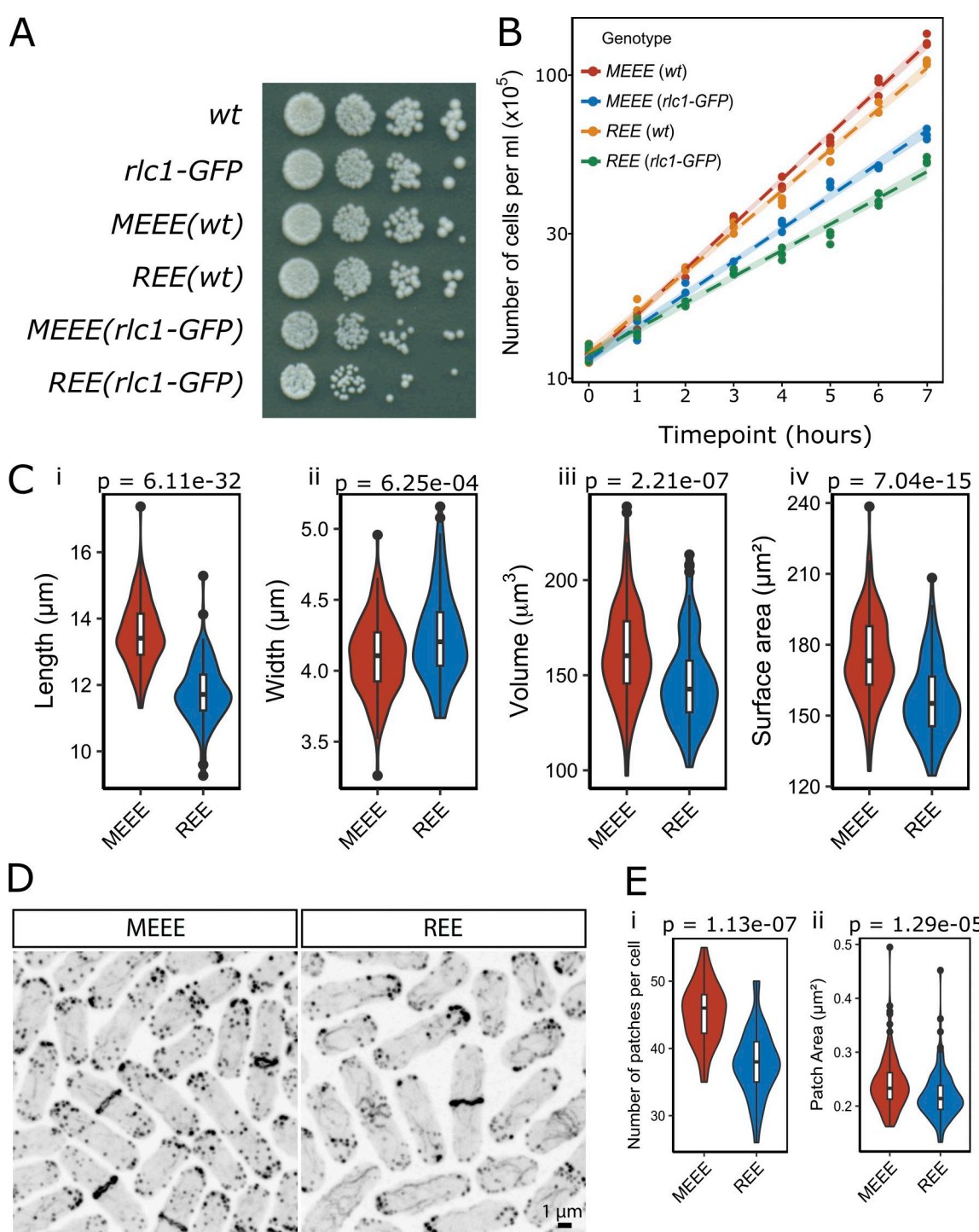

Figure 4. **Viability, growth, cell size, and actin distribution in the *REEact1* and control *S. pombe* strains. (A)** Spot assay showing the growth of 10-fold serial dilutions of the indicated strains that were spotted on YEA plates and grown at 30°C for 3 days. **(B)** Growth of the four strains fitted to a linear regression model with 95% confidence intervals and placed on a $\log_{10}$ Y axis scale. The dots show the cell counts. **(C)** Violin plots of the cell size parameters of (i) Length, (ii) Width, (iii) Volume and (iv) Surface Area for the two indicated strains in the *wt* background (in total 143 and 122 septated cells from 3 sets were measured for the MEEE and REE strains, respectively. Data is mean ± SD; length (µm): MEEE 13.54 ± 0.98, REE: 11.76 ± 0.90, Wilcox test: P < 6.11e-32; width (µm): MEEE 4.10 ± 0.27, REE: 4.24 ± 0.31, Wilcox test: P = 6.25e-04; volume (µm³): MEEE 161.32 ± 24.98, REE: 146.49 ± 23.35, Wilcox test: P = 2.21e-07; surface area (µm²): MEEE 174.45 ± 18.05, REE: 156.56 ± 16.14, Wilcox test: P = 7.04e-15). **(D)** Phalloidin stainings of the indicated strains in the *wt* background. **(E)** Violin plots of the number of fimbrin-GFP patches (i) and patch area at the brightest slice (ii) (data written as n; mean ± SD; counts: MEEE: 34 cells; 45.24 ± 4.42; REE: 37 cells from 3 different experiments; 37.86 ± 5.08; Wilcox test: P = 1.13e-07; area (µm²): MEEE: 144 patches; 0.24 ± 0.05; REE: 144 patches; 0.22 ± 0.04; Wilcox test: P = 1.29e-05;). Scale bars in D are 1 µm.

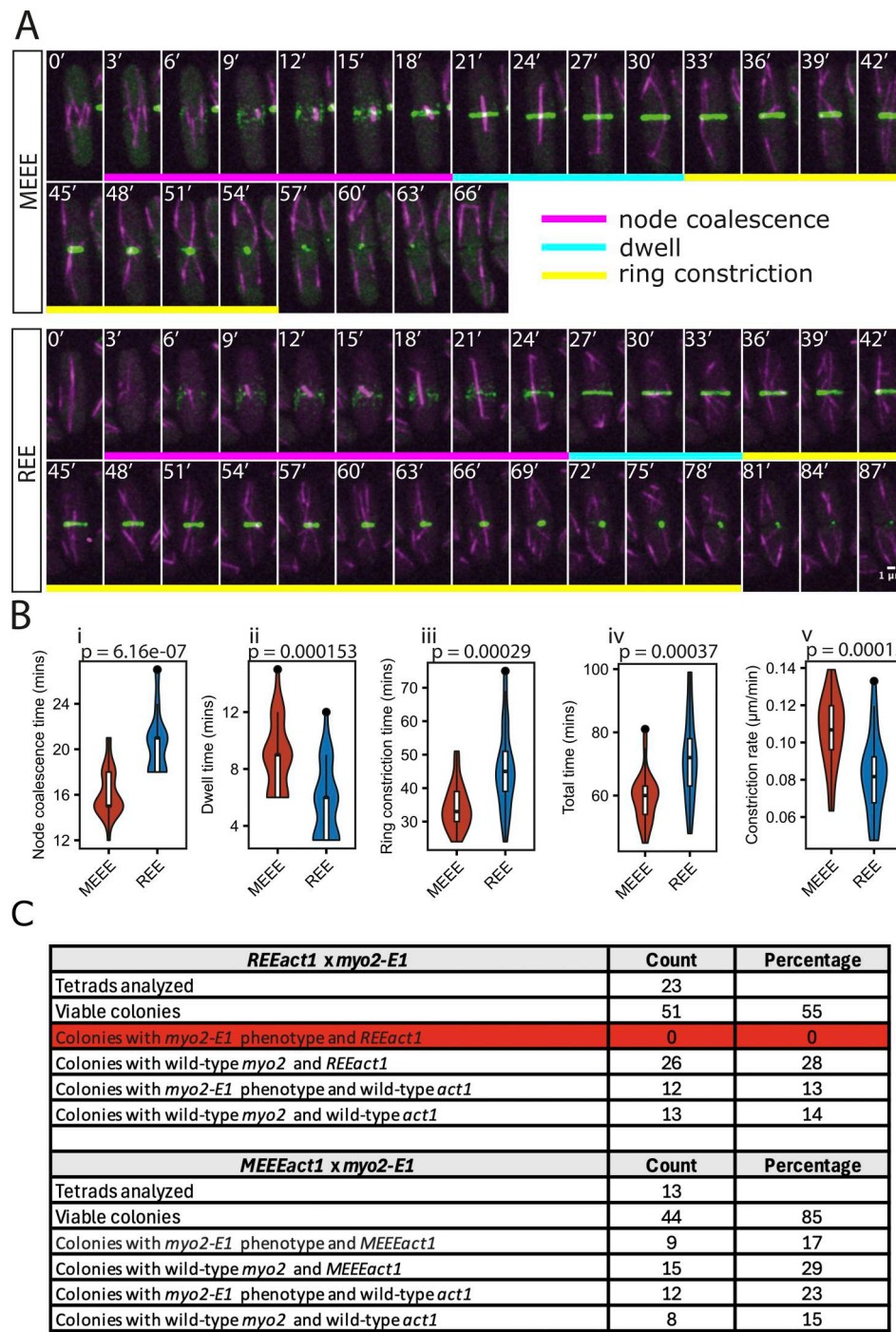

Figure 5. **Cytokinesis in the *REEact1* and control *S. pombe* strains and tetrad analysis when crossed to the *myo2*-E1 strain. (A)** Representative montages of time-lapse movies of live cells from the indicated strains in the *rlc1-GFP* background showing mCherry-Atb2 (magenta) and Rlc1-3GFP (green). Images were captured at 3-min intervals. **(B)** Violin plots of the analysis of ring constriction (*n* = 25 cells each) for node coalescence (data written as n; mean ± SD; REE: 20.52 min ± 2.69; MEE: 16.20 ± 2.12; Wilcox test P = 6.16e-07) (i), dwell time (REE: 5.64 min ± 2.64; MEEE: 8.88 min ± 2.52; Wilcox test P = 0.000153) (ii), ring constriction time (REE: 45.48 min ± 12.27; MEEE: 34.32 min ± 6.66; T test P = 0.00029) (iii), total time (REE: 71.64 min ± 13.54; MEEE: 59.40 min ± 7.94; T test P = 0.00037) (iv) and constriction rate (REE: 0.083 μm/min ± 0.021; MEEE: 0.106 μm/min ± 0.019; *T* test P = 0.00015) (v). **(C)** Table of counts of the genotypes of colonies arising out of the tetrad dissection of asci from crosses of *myo2*-E1 and either REE or MEEE strains. Scale bars in A are 1 μm.

the absence of ATP). Interestingly, when both actin isoforms were mixed 1:1, no difference in filament landing to the surface and no difference in filament motility was detected compared with Ac-β-actin samples only (data not shown). However, R-β-actin concentration in different parts of cells may vary.

Filaments containing mostly R-β-actin might be more common on the leading edge of the cell facilitating migration of the cell (Pavlyk et al., 2018).

Most previous work on R-actin in cells use knockout or inhibition of the arginyltransferase enzyme *ate1*, which arginylates

actin (Karakozova et al., 2006; Saha et al., 2010; Saha et al., 2012). A limitation is that ATE1 also arginylates many other molecules, making it difficult to determine if the observed phenotypes result specifically from the loss of R-actin or other arginylated targets. Our results demonstrate the functional difference between Ac-actin and R-actin in the cell. One question that has not been resolved to date is if R-actin can perform all the functions of wt actin in cells or if it can only interact with a subset of actin-binding proteins. Since actin's interactions with the Arp2/3 complex, formins, and myosin are critical to cell function, it is reasonable to assume that significantly compromised functional interactions with any of these proteins and actin will be deleterious or even lethal to cells. Interestingly, we found that cells were capable of survival with subtle actin patch, cable, and cytokinesis defects when REE-actin was the sole genetically provided actin in the cell. Actin patches are nucleated by the Arp2/3 complex (Kaksonen et al., 2003), suggesting that Arp2/3-based actin nucleation is likely compromised in the REE strains. This is consistent with defective Arp2/3 complex-based actin nucleation of R-β-actin, which we observed previously (Chin et al., 2022). Thicker cables are an indicator of failed myosin function or an anticorrelation between patches and cables in fission yeast (Balasubramanian et al., 1996; Balasubramanian et al., 1998; Burke et al., 2014; Zambon et al., 2020). It is very likely, based on our *in vitro* analysis, that the charge differences in REE-actin and its consequent decrease in myosin II interaction led to the cytokinesis defects that we observed in the REE strain. It should be noted that experiments in fission yeast show arginylated actin can perform all functions needed for a viable cell, but this may not reflect the situation in mammalian cells. *S. pombe* lacks actin arginylation and actin isoforms, so its role in cells where most actin is acetylated, like mammalian cells, remains unclear. While *S. pombe* has Arp2/3 and formin actin structures, it lacks lamellipodia, cell adhesions, and stress fibers typical of mammalian cells. Our work shows arginylation does not change actin's overall structure, but the charge change at the N terminus likely affects local interactions, influencing its localization and function.

## Materials and methods
### Experimental strains
#### Yeast strains, media, and culture conditions
Fission yeast *S. pombe* strains used in this study have been listed below. Cells used for live imaging or staining were grown in YEA medium at 30°C with shaking at 200 rpm. The *MEEEact1* and *REEact1* strains were generated using the SpEDIT system (Torres-Garcia et al., 2020). Briefly, an sg-RNA–loaded pLSB-Nat plasmid (a gift from Robin Allshire, University of Edinburgh, Edinburgh, UK, #166698; Addgene plasmid; http://n2t.net/addgene:166698; RRID:166698; Addgene) was generated using the primers in a golden gate cloning reaction:

(F:5′-CTAGAGGTCTCGGACTACCCTCAAAAGACAAGACCA GTTTCGAGACCCTTCC-3′ R: 5′-GGAAGGGTCTCGAAACTGGTC TTGTCTTTTGAGGGTAGTCCGAGACCTCTAG-3′).

Targeting the 5′UTR near the ATG site of *act1*. A homologous recombination template (HDR-template) containing the *ubi4-*

*MEEE-act1* (till the stop codon) or *ubi4-REE-act1* (till the stop codon) was generated using PCR from plasmids containing these sequences using the following primers and Phusion polymerase (New England Biolabs: M0530S):

(F:5′-AATCAACGGCTTCATACCACCTCAGCCAGCCGTGTT ATAACTTACCGTTTACCAACTACATTTTTTGTAACGAACCAA AAAACCCTCAAAAGACAAGACCATGCAGATTTTCGTCAAGAC-3′ R: 5′-TTAGAAGCACTTACGGTAAACG-3′).

The products were PCR-purified using a Qiagen PCR purification kit (28104). The HDR-template and sgRNA-pLSB plasmid were transformed into the *wt* and *rlc1-GFP* strains, and colonies were selected on YEA plates containing NAT antibiotic at 24°C. The colonies were then transferred to YEA plates and screened using PCR to detect Ubi4 incorporation using the following genotyping primers:

(F:5′-GCATTCTGCCGTGAAGTG-3′ R: 5′-GCTCAAAGTCCA AAGCGAC-3′).

They were then sequenced using the service from GATC, and the sequences were aligned using SnapGene software (from Dotmatics; available at snapgene.com, RRID:SCR_015052), and confirmed. Schematics related to DNA sequences were also made in SnapGene.

**Strains.** The strains used are listed below in the format strain, source, lab collection number; *h+* (referred to as wt); laboratory collection, *MBY100; m-cherry Atb2::hph Rlc 3XGFP::kan Mx ura4 -D18* (referred to as *rlc1-GFP*); laboratory collection, *MBY5985; MBY100 + Ubi4-MEEE_act1* at the native *Act1* locus; this study, *MBY13727; MBY100 + Ubi4-REE_act1* at the native *Act1* locus; this study, *MBY13728; MBY5985 + Ubi4-MEEE_act1* at the native Act1 locus; this study, *MBY13729; MBY5985 + Ubi4-REE_act1* at the native *Act1* locus; this study, *MBY13730; 12601* with actin: *AOX1<pPICZc-Ub-R-HsACTB-thymosinB-8His* (#TH4-25), *his4-<pIB2-SETD3* (#TH6-81) *P. pastoris*, (Hatano et al., 2018), *MBY12817; myo2-E1 mCherry atb2::hph rlc1-3GFP::kan Mx h-*; laboratory collection, *MBY10024-4; FimbrinGFP::KanMX6 REE-Act1 leu1-32*; this study, *MBY14325; FimbrinGFP::KanMX6 MEEE-Act1 leu1-32*; this study, *MBY14326*.

### Imaging and image analysis of S. pombe strains
**Spinning disk confocal microscopy.** Imaging was performed using the Andor Revolution XD spinning disk or the Andor Tu-Cam spinning disk confocal microscope. Imaging was performed at 30°C for all live-imaging experiments. The spinning-disk confocal system consisted of a Nikon ECLIPSE Ti inverted microscope with a Nikon Plan Apo Lambda 100×/1.45 NA oil immersion objective lens, a spinning-disk system (CSU-X1; Yokogawa), and an Andor iXon Ultra EMCCD camera. A pixel size of 80 nm/pixel for the Andor Revolution or 69 nm/pixel for the Andor TuCam was used for acquiring the images using the Andor IQ3 software. Lasers with wavelengths of 488 nm or 561 nm were used to excite the fluorophores. The time-lapse images were acquired with a Z-step of 500 nm.

**Preparation of cells for live imaging.** Cells used for live imaging were grown in YEA medium at 30°C with shaking at 200 rpm and grown to mid-log phase. 1 ml of cells was centrifuged at 450 × *g* for 2 min to concentrate them to 20–100 µl and mounted sandwiched between an agarose pad and a coverslip,

which was sealed with valap (a mixture of Vaseline, lanolin, and paraffin), and imaged. Images were collected at 3-min intervals.

**Phalloidin staining of S. pombe strains.** For phalloidin staining, the cells were grown in YEA medium at 30°C with shaking at 200 rpm. They were grown to mid-log phase. 50 ml of the culture was spun down at 1900 $g$ for 1 min, washed once with PBS, and spun again. 500 µl of PBS was added to the pellet, and 500 µl of the resultant solution was taken for fixing. 500 µl of 8% PFA was added to the 500 µl of cells and mixed immediately and then shaken on a nutator for 45 min. The cells were spun down at 845 $g$ for 1 min, and the PFA was discarded. The cells were washed once with PBS and then kept in 1 ml PBS at 4°C till processing. 100 µl of these cells were taken, and 1 ml of 1% PBT was added to them for 30 min at RT. The cells were washed once with PBS, and the pellet was resuspended in 10 µl of 1:10 phalloidin (R415; Molecular probes) and 1 µg/ml DAPI (MBD0015; Merck) in PBS and mounted between an agarose pad and coverslip for imaging.

**Analysis of S. pombe CAR-related parameters.** Live images were analyzed manually in FIJI (RRID:SCR_002285) by CP for ring-related parameters (Schindelin et al., 2012). Statistics were computed and plotted using R (RRID:SCR_001905) in Rstudio (RRID:SCR_000432) (RCoreTeam, 2022; RStudioTeam, 2018).

**Quantification of S. pombe patches.** For assessment of patches, fimbrin-GFP cells that were imaged live at a single time point at the mid-log phase were used to manually count the number of patches per cell in 3D in Fiji using the multipoint tool. For measuring patch area, well-separated patches were chosen and the area of the patch was obtained in FIJI using the polygon tool at the slice with the visually brightest patch intensity.

**Analysis of S. pombe cell size parameters.** For measuring cell size, the cells were grown in YEA at 30°C to mid-log phase. The cells were then imaged live on the Andor TuCam with DIC. Cell images were analyzed manually in Fiji using the line tool to draw the length and width in septated cells. Volume and surface area were calculated from these values as follows: Volume = $\pi(width/2)^2(length-width)+ 4/3*\pi(width/2)^3$ and surface area = $2\pi(width/2)(length-width)+4\pi(width/2)^2$.

## Western blotting of S. pombe cells
The four strains were grown overnight in YEA broth. 33 ml of 0.3 OD cells were then then lysed using glass beads and vortexing for 1 min and stored on ice and extracted with 500 µl of PBS; 300 µl of this solution was added to 100 µl of 4× Laemelli buffer and was heated at 95°C for 5 min. They were then loaded and run on 10% SDS-PAGE gels, followed by western blotting using the BIO-RAD turbo-blot using the preset mini gel setting. The blot was then blocked with 10% skimmed milk powder in TBS-Tween, followed by primary anti-actin antibody in blocking overnight at 4°C (MERCK: MAB1501 RRID:AB_2223041 at 1:1,000). The blots were washed four times with TBS-Tween for 10 min, followed by anti-mouse HRP (7076S; Cell signaling at 1:3,000) for 1 h at room temperature.

## Mass spectroscopy
### Sample preparation
The strains were prepared for SDS-PAGE as above for western blotting. The SDS-PAGE gel was stained with Coomassie blue, and bands between 35 and 50 kDa were cut. The samples were reduced using TCEP and alkylated. They were then digested overnight using trypsin and concentrated using a speed-vac to 20 µl.

### NanoLC-ESI-MS/MS analysis
Reversed phase chromatography was used to separate tryptic peptides prior to mass spectrometric analysis. Two columns were utilized, an Acclaim PepMap µ-precolumn cartridge 300 µm i.d. × 5 mm 5 µm 100 Å and an Acclaim PepMap RSLC 75 µm × 50 cm 2 µm 100 Å (Thermo Fisher Scientific). The columns were installed on an Ultimate 3000 RSLCnano system (Thermo Fisher Scientific). Mobile phase buffer A was composed of 0.1% formic acid in water, and mobile phase B was 0.1% formic acid in acetonitrile. Samples were loaded onto the µ-precolumn equilibrated in 2% aqueous acetonitrile containing 0.1% trifluoroacetic acid for 5 min at 10 µl min-1, after which peptides were eluted onto the analytical column at 250 nl min-1 by increasing the mobile phase B concentration from 4% B to 25% over 36 min, then to 35% B over 10 min, and to 90% B over 3 min, followed by a 10 min re-equilibration at 4% B.

Eluting peptides were converted to gas-phase ions using electrospray ionization and analyzed on a Thermo Orbitrap Fusion (Q-OT-qIT, Thermo Fisher Scientific) (Hu et al., 2005). Survey scans of peptide precursors from 375 to 1575 m/z were performed at 120 K resolution (at 200 m/z) with a 50% normalized AGC target, and the max injection time was 150 ms. Tandem MS was performed by isolation at 1.2 Th using the quadrupole, HCD fragmentation with normalized collision energy of 33, and rapid scan MS analysis in the ion trap. The MS2 was set to 50% normalized AGC target, and the max injection time was 200 ms. Precursors with charge state 2–6 were selected and sampled for MS2. The dynamic exclusion duration was set to 45 s with a 10 ppm tolerance around the selected precursor and its isotopes. Monoisotopic precursor selection was turned on. The instrument was run in top speed mode with 2-s cycles.

### Data analysis
The data were then analyzed using Mascot (Perkins et al., 1999) using the following parameters (enzyme: trypsin; fixed modifications: carbamidomethyl (C); variable modifications: oxidation (M), acetyl (N-term; mass values: monoisotopic; protein mass: unrestricted; peptide mass tolerance: ± 10 ppm; fragment mass tolerance: ± 0.6 Da; max missed cleavages: 3).

## Spot dilution assay
The different strains were grown in YEA broth overnight at 30°C. The following day they were diluted and grown to mid-log phase. The cells were then diluted to 0.1 OD (595 nm), 0.01, 0.001, and 0.0001 OD on YEA agar plates and incubated at 30°C for 3 days. They were then scanned on an EPSON perfection V700 Photo scanner.

## Analysis of S. pombe growth
For measuring growth, the cells were grown in YEA at 30°C overnight. The following day, the cultures were mid-log phase. These cultures were diluted to 0.1 OD and grown in conical flasks with shaking in prewarmed YEA. Every hour, an OD reading was

taken, and 750 µl of cells were fixed with 250 µl of 16% formaldehyde with shaking for 1 h and then stored at 4°C till cell counting. The cells were washed thrice in PBS, taking care that the volumes remained unchanged. Three replicates were performed. Cell counting was performed using a Neubauer Haemocytometer. The script for fitting was developed in R using vibe coding with ChatGPT. Cell counts were fitted to a log-transformed linear regression model. From each model, fitted values and 95% confidence intervals were computed over a sequence of 100 evenly spaced time points spanning the observed range. These predictions were back-transformed to the original scale and plotted in R.

## Actin purification

Actin was purified as previously described (Hatano et al., 2020). Briefly, MBY12817 cells were grown in MGY liquid medium composed of 1.34% yeast nitrogen base without amino acids (Y0626; Sigma-Aldrich), 0.4 mg/l biotin, and 1% glycerol at 30°C, 220 rpm to an OD600 of 1.5. They were then pelleted by centrifugation and washed with sterile water. They were then cultured in MM medium containing 1.34% yeast nitrogen base without amino acids (Y0626; Sigma-Aldrich), 0.4 mg/l biotin, and 0.5% methanol at 30°C and 220 rpm for 1.5–2 days. The cells were pelleted, washed, and a frozen in liquid nitrogen. The frozen cells were lysed using a cryo mill (#6870; SPEX SamplePrep). The lysate was resuspended in an equal volume of 2× binding buffer composed of 20 mM imidazole (pH 7.4), 20 mM HEPES (pH 7.4), 600 mM NaCl, 4 mM MgCl$_2$, 2 mM ATP (pH 7.0), 2× concentration of protease inhibitor cocktail (cOmplete, EDTA free #05056489001; Roche), 1 mM PMSF, and 7 mM beta-mercaptoethanol. The lysate was sonicated (5 s with 60% amplitude, Qsonica Sonicators). The lysate was then centrifuged at 25,658 g for 5 min to remove cell debris and a further 1 h to remove the insoluble fraction. The lysate was then clarified using a 0.22-µm filter. It was then incubated with Ni-NTA beads (#88222; Thermo Fisher Scientific) at 4°C for 1 h. The resin was pelleted and washed repeatedly with 1× binding buffer and then G-buffer containing 5 mM HEPES (pH 7.4), 0.2 mM CaCl$_2$, 0.01 wt/vol% NaN$_3$, 0.2 mM ATP (pH 7.0), and 0.5 mM DTT. Actin was then cleaved off the beads overnight at 4°C using 5 µg/ml TLCK-treated chymotrypsin (#C3142-25MG; Sigma-Aldrich), which was then inactivated by 1 mM PMSF. The beads were pelleted, and the supernatant was concentrated using a 30-kDa cutoff membrane to 0.9 ml. The actin was polymerized using 100 µl of 10× MKE solution composed of 20 mM MgCl$_2$, 50 mM EGTA, and 1 M KCl for 1 h at room temperature and pelleted down by ultracentrifugation at room temperature (124500 g for 1 h, Beckman TLA-55 rotor). The pellet was then resuspended in 1× G-buffer and dialyzed against 2 liters of G-buffer over 48 h at 4°C.

Actin for EM was polymerized by mixing 20 µl of 20 µM G-actin, 8 µl of 10× MKE, and 52 µl of 5 mM HEPES-KOH, pH 7.4, containing 0.2 mM ATP and 0.5 mM DTT, and incubating at RT for 1 h.

## Myosin purification
### Tissue purified porcine cardiac myosin
Porcine cardiac myosin for motility assays was purified as described previously, (Murakami et al., 1976). The protocol was not modified. The myosin was stored in 50% glycerol at –20°C.

### NM myosin II A
HMM NMIIA was recombinantly produced in the baculovirus/Sf9 insect cell system and prepared to homogeneity via FLAG affinity purification as described before (Wang et al., 2000) with minor modifications. For purification, cells were lysed by sonication in buffer containing 10 mM MOPS, pH 7.3, 0.2 M NaCl, 2 mM ATP, 10 mM MgCl$_2$, and 1 mM EGTA, supplemented with protease inhibitors and centrifuged (41,856 g, 35 min, 4°C). The supernatant was incubated with anti-FLAG affinity resin, washed with buffer containing 10 mM MOPS, pH 7.3, 500 mM NaCl, 0.1 mM EGTA, 1 mM ATP, and 5 mM MgCl$_2$, followed by a wash in the same buffer without ATP and MgCl$_2$. The protein was eluted in buffer containing 10 mM MOPS, pH 7.3, 150 mM NaCl, 0.1 mM EGTA, and 100 µg/ml FLAG peptide. The protein was dialyzed overnight against 10 mM MOPS, pH 7.3, 500 mM NaCl, and 0.1 mM EGTA and further purified on a HiLoad 16/600 Superdex 200-pg column. The protein was concentrated by ultrafiltration (MWCO 50 kDa), flash frozen, and stored at –80°C.

### Myosin light chain kinase
Rabbit smooth muscle myosin light chain kinase (MLCK, NP_001075775) was produced in the baculovirus/Sf9 insect cell system and prepared to homogeneity via FLAG affinity purification as described. In brief, cells were lysed in buffer containing 10 mM MOPS, pH 7.3, 200 mM NaCl, 1 mM EGTA, and 0.1 mM DTT supplemented with protease inhibitors and centrifuged (41,856 g, 35 min, 4°C). The supernatant was incubated with anti-FLAG affinity resin, washed with buffer containing 10 mM MOPS, pH 7.3, 500 mM NaCl, and 0.1 mM EGTA and eluted in buffer containing 10 mM MOPS, pH 7.3, 150 mM NaCl, 0.1 mM EGTA, and 100 µg/ml FLAG peptide. The protein was dialyzed overnight in buffer containing 10 mM MOPS, pH 7.3, 150 mM NaCl, 0.1 mM EGTA, and 2 mM DTT, concentrated by ultrafiltration (MWCO 30 kDa), flash frozen, and stored at –80°C.

## Actin-activated myosin ATPase assay
Actin-activated myosin ATPase activity assay was performed using EnzChek ATPase kit (Thermo Fisher Scientific). The assay was performed in a 384-well plate at 21°C. Reaction buffer (5 mM KCl, 0.6 mM MgCl2, and 10 mM PIPES) was mixed with 2-amino-6-mercapto-7-methylpurine riboside, purine nucleoside phosphorylase, polymerized actin, human NM HMM2A, and ATP (40 mM). As soon as actin, myosin, and ATP were mixed with other components, the absorbance of the assay mixture was measured at 360 nm for 1 h. Then rate of absorbance change was calculated for all conditions.

## Motility assay
### Sample preparation and imaging
The movement of actin filaments on myosin was assessed using a motility assay. Acetylated actin or arginylated actin was copolymerized with 5% fluorescently labelled Ac-actin with Alexa Fluor 488 (Invitrogen) or pre-stained with phalloidin Alexa 488 (8878S; Cell Signaling) after polymerization. Actin was polymerized at room temperature for 1 h by adding MEK polymerization buffer (20 mM MgCl2, 50 mM EGTA, and 1 M KCl). Glass coverslips were cleaned in 2% Hellmanex II for at 60°C and then in 3 M NaOH for 30 min at 60°C. Both steps were performed in

ultrasonic bath. Cleaned coverslips were rinsed with pure water and air-dried. Then coverslips were coated with silicone by immersing them in 5% (vol/vol) Sigmacote (Sigma-Aldrich) in heptane or with 1% nitrocellulose (Sigma-Aldrich). The coverslips were attached to microscopy slides using three stripes of double-sided sellotape. Each coverslip had two 2-mm wide channels. Porcine cardiac myosin (1 mg/ml) was loaded into both channels and incubated for 10 min at room temperature. Then the channels were washed with 20 µl running buffer (25 mM KCl, 4 mM MgCl2, 1 mM ATP, 10 mM DTT, 25 mM Imidazole, pH 7.4, and 1 mg/ml BSA) twice. The channels were incubated with running buffer for 2 min. 20 µl of running buffer supplemented with 20–40 nM of polymerized actin were loaded into channels. Similarly, HMM NM myosin II (140 µg/ml) was loaded into both channels and incubated for 10 min at room temperature. Then the channels were washed with 10 µl washing buffer (25 imidazole, 25 mM KCl, 4 mM Mg$_2$Cl, 1 mM ATP, and 5 mM DTT). 5 µl of running buffer (25 imidazole, 25 mM KCl, 4 mM Mg$_2$Cl, 7.6 mM ATP, 50 mM DTT, and 0.5% methyl cellulose) and supplemented with 25 nM of polymerized actin were loaded into channels.

The first channel was loaded with acetylated actin, while second with arginylated actin. The channels were sealed with nail polish. The samples were imaged with confocal fluorescence microscope. Images were taken every 100 msec for 20 s when cardiac myosin was used and every 1 s for 1 min when NM myosin was used. Actin filament movements apart from detachment and wobbling were analyzed with Fiji and TrackMate plugin or a python (RRID:SCR_008394) script written for this assay using vibe coding with ChatGPT.

### Image analysis
The pipeline for image analysis of detachment and wobbling of actin filaments during motility is illustrated in Fig. S2 D. To visualize detachment and wobbling, Gaussian blurring with a radius of two pixels was first applied to the images. The ratios between the pixel intensities of the $n$th and $(n+1)$-th images of the movie were then calculated. These ratio images were maximum-projected through the time points and presented on a linear scale from 1.3 to 3 using the pseudocolor "FIRE" lookup table. For scoring the detachment and wobbling events, the actin filaments and background in the original movie were first segmented using Trainable Weka Segmentation (Arganda-Carreras et al., 2017) in Fiji. The mean and SD of the background pixel intensities were measured. Spots in the ratio images that exceeded a threshold defined as

$$1 + 4\left(\frac{\text{SD of the background}}{\text{mean of the background}}\right)$$

were counted as detachment or wobbling events.

### EM
Samples were prepared by applying the polymerized R-β-Actin actin to freshly glow-discharged Quantifoil 3.5/1 on 200 mesh copper grids (Quantifoil GmbH), using a Leica GP2 (Leica Microsystems GmbH). Globular R-β-actin was polymerized, plunge-frozen, and screened on a JEOL2200FS with a K2 camera to evaluate filament formation and ice thickness.

Data collection was carried out on the Thermo Fisher Scientific Titan Krios using the Gatan K2 Direct Electron Detector,

operated at a nominal magnification of 75,000×, resulting in a calibrated pixel size of 1.08 Å/pix (Table S1). Micrographs were recorded as movies comprising 39 individual frames recorded over 60 s and with a dose rate of 0.7 electrons per pixel per second, giving a total dose per image of 42 electrons/pixel.

### Image processing and model building
All data processing was done using RELION3.0.9 (RRID:SCR_016274) (Zivanov et al., 2018). Image stacks were motion-corrected and summed using MotionCor2 (RRID:SCR_016499) (Zheng et al., 2017), and CTF parameters were calculated using Gctf (RRID:SCR_016500) (Zhang, 2016). Initially, filament segments were manually selected from a subset of micrographs, and 2D classes were calculated. These were then used to optimize the settings for autopicking on a subset of micrographs with a range of defocus values, before running autopicking on all images. Bad particles were removed using 2D and 3D classification before refining the helical parameters in 3D refinement.

Global resolution was estimated using the Fourier shell correlation 0.143 criterion, and local resolution maps were produced using LocResMap.

PDB model 3J8I (Galkin et al., 2015) was fitted into the density map to assess similarity to previous structures. Model building was performed in COOT (RRID:SCR_014222) (Emsley and Cowtan, 2004), mutating the primary sequence to account for the difference between chicken actin and human actin. Refinements were carried out in reciprocal space using REFMAC (RRID:SCR_014225) (Murshudov et al., 1997), and geometry was analyzed using Coot and MolProbity (RRID:SCR_014226) (Chen et al., 2010).

### Data deposition
The map for R-β-actin has been deposited in the EM data bank—accession number EMD-16776—and in the protein data bank with the following accession code PDB ID 8COG.

### A model of actomyosin complex
An actin–myosin 1:1 complex was modelled based on chains A and F of a cryo-EM structure (PDB:5JLH [von der Ecken et al., 2016]). The missing loops of NM myosin IIC were modelled with MODELLER (RRID:SCR_008395) (Sali and Blundell, 1993) via UCSF Chimera (RRID:SCR_015872) (Pettersen et al., 2004). Molecular dynamic simulation was performed with GROMACS (https://www.gromacs.org/ RRID:SCR_014565 [Páll et al., 2020]) using the output from CHARMM-GUI (https://www.charmm-gui.org/ RRID:SCR_025037 [Lee et al., 2016]) with N-terminal acetylation of the gamma actin. Coulombic surface coloring was done by UCSF Chimera.

ChatGPT was used to reduce word count in two occasions.

## Online supplemental material
Fig. S1 shows the local resolution map and comparison of the structures. Fig. S2 shows data and algorithm related to the gliding assays and analysis. Fig. S3 shows how the *S. pombe* strains were generated. Fig. S4 shows the validation of the *S. pombe* strains. Table S1 shows the cryo-EM–related statistics. Video 1 shows the gliding assays. Video 2 shows ring constriction in the *S. pombe* strains.

**Data availability**

The data are available upon reasonable request.

## Acknowledgments

We acknowledge the Warwick Advanced Bioimaging Research Technology Platform supported by BBSRCBB/M01228X/1 and the Midlands Regional Cryo- EM Facility (Leicester) for use of the Titan Krios, supported by MRC grant MC_PC_17136. We thank TJ Ragan for his assistance with data collection and the Warwick Proteomics Core facility for mass spectrometry.

This work was supported by Wellcome Trust (WT101885MA; Mohan Balasubramanian) and (203276/Z/16/Z; Mohan Balasubramanian); European Research Council (ERC-2014-ADG No. 671083; Mohan Balasubramanian); HFSP (RGP-001-2023); and BBSRC (BB/S003789/1; Mohan Balasubramanian, Masanori Mishima, and Karuna Sampath), (National Institutes of Health [NIH] R01GM143414; Sarah M. Heissler), and (NIH R01GM143539; Krishna Chinthalapudi).

Author contributions: Clyde Savio Pinto: conceptualization, data curation, formal analysis, investigation, methodology, software, validation, visualization, and writing—original draft, review, and editing. Saskia E. Bakker: formal analysis, investigation, methodology, validation, visualization, and writing—original draft, review, and editing. Andrejus Suchenko: data curation, formal analysis, investigation, methodology, software, validation, visualization, and writing—original draft, review, and editing. Isabella M. Kolodny: investigation. Hamdi Hussain: conceptualization, formal analysis, methodology, project administration, resources, visualization, and writing—original draft, review, and editing. Tomoyuki Hatano: investigation and resources. Karuna Sampath: conceptualization, funding acquisition, supervision, and writing—review and editing. Krishna Chinthalapudi: data curation, formal analysis, methodology, and writing—original draft, review, and editing. Sarah M. Heissler: funding acquisition, methodology, resources, supervision, and writing—review and editing. Masanori Mishima: conceptualization, data curation, formal analysis, funding acquisition, methodology, project administration, software, supervision, visualization, and writing—review and editing. Mohan Balasubramanian: conceptualization, funding acquisition, project administration, supervision, and writing—original draft, review, and editing. Disclosures: The authors declare no competing interests exist.

Submitted: 12 September 2024

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

# Supplemental material

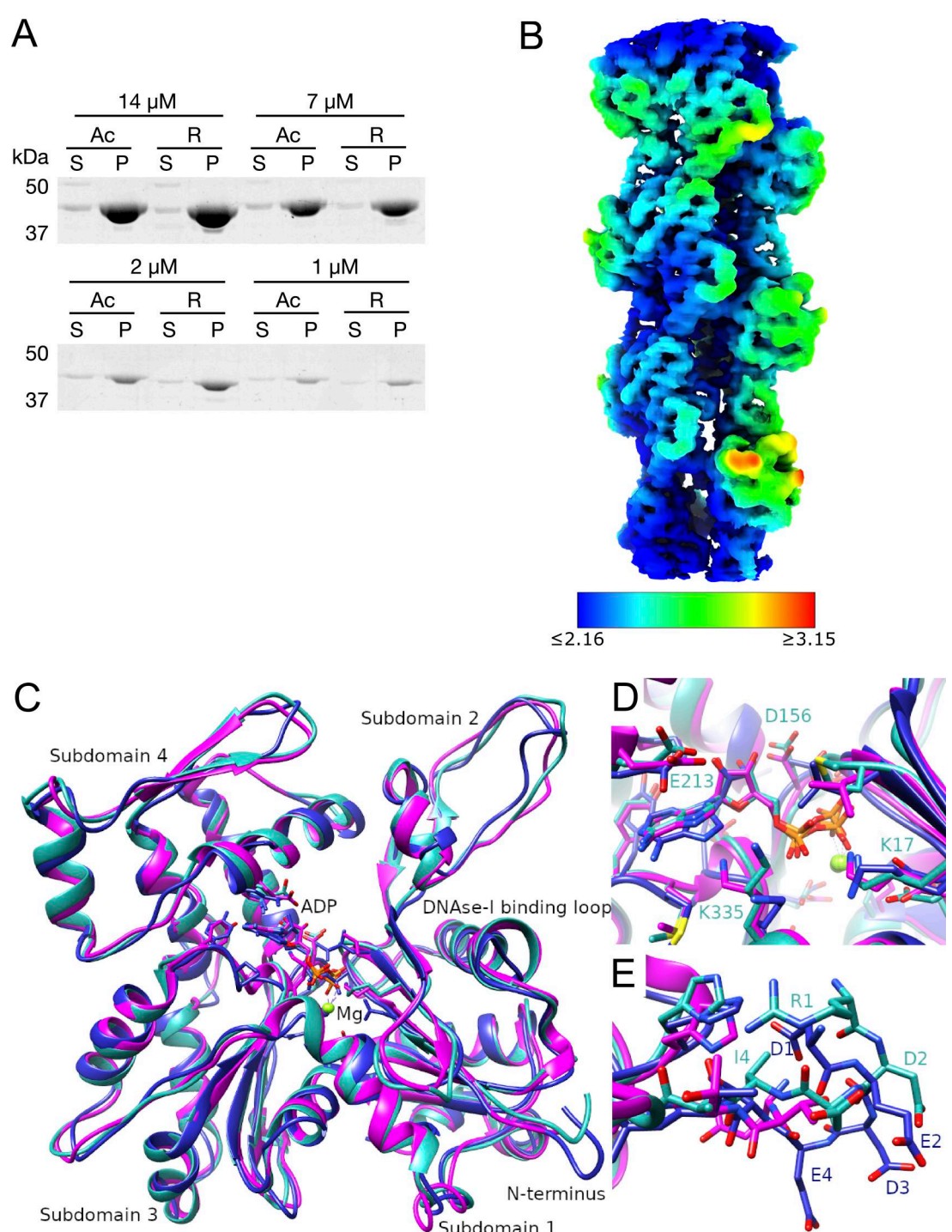

Figure S1. **The local resolution map of R-β-actin and comparison of the R-β-actin, chicken, and rabbit skeletal muscle actin structures. (A)** The purified Ac- and R-actin were polymerized *in vitro* and ultracentrifuged at 100,000 × *g* for 1 h. The proteins in the supernatant (S) and pellet (P) were assessed by SDS-PAGE and Coomassie staining. **(B)** The local resolution map of R-β-actin color coded as per the legend. **(C)** Ribbon diagram comparing the structure of R-β-actin presented here (cyan) with previously published structures of chicken (navy) and rabbit (magenta) skeletal muscle actin. **(D)** Close-up view comparing the structure of the active site of R-β-actin with previously published structures of chicken and rabbit skeletal muscle actin. **(E)** Close up view comparing the N-terminal sections of R-β-actin to that of chicken and rabbit skeletal muscle actin. Source data are available for this figure: SourceData FS1.

JCB

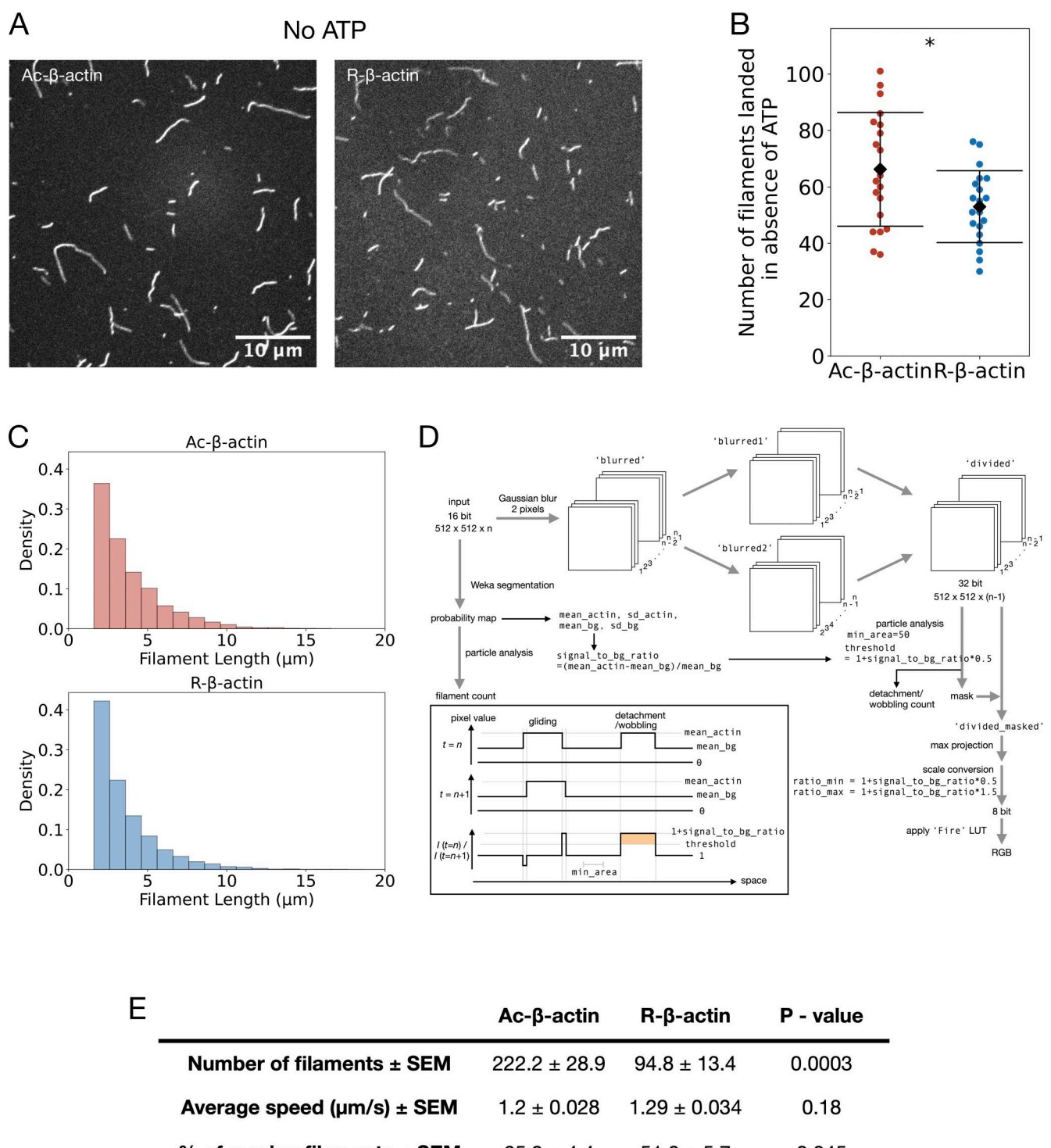

**Figure S2.** **The degree of assembly into filaments and the length distribution of the arginylated actin filaments. (A)** Phalloidin Alexa Fluor-488–labeled Ac- and R-actin filaments were loaded on the surfaced coated with NM myosin II in the absence of ATP. **(B)** The number of the landed filaments per movie. **(C)** The length distributions of the landed filaments. **(D)** Schematic of the image analysis pipeline to detect and analyze detachment/ wobbling of actin filaments in the surface motility assay. The pixels that showed a large difference from the previous time frame were detected by dividing the (i+1)-th images with the *i*th images after smoothing with Gaussian blurring. The maximum projections through the time frames are shown in a pseudocolor, as shown in Fig. 3D. For quantification of the detachment/wobbling events, a threshold was calculated based on the mean and the SD of the background areas in each movie, and the spots above this threshold value in the (i+1)-th/*i*th ratio movie were detected. **(E)** Actin-gliding data obtained from analysis of motility assay performed with tissue isolated full-length cardiac myosin.

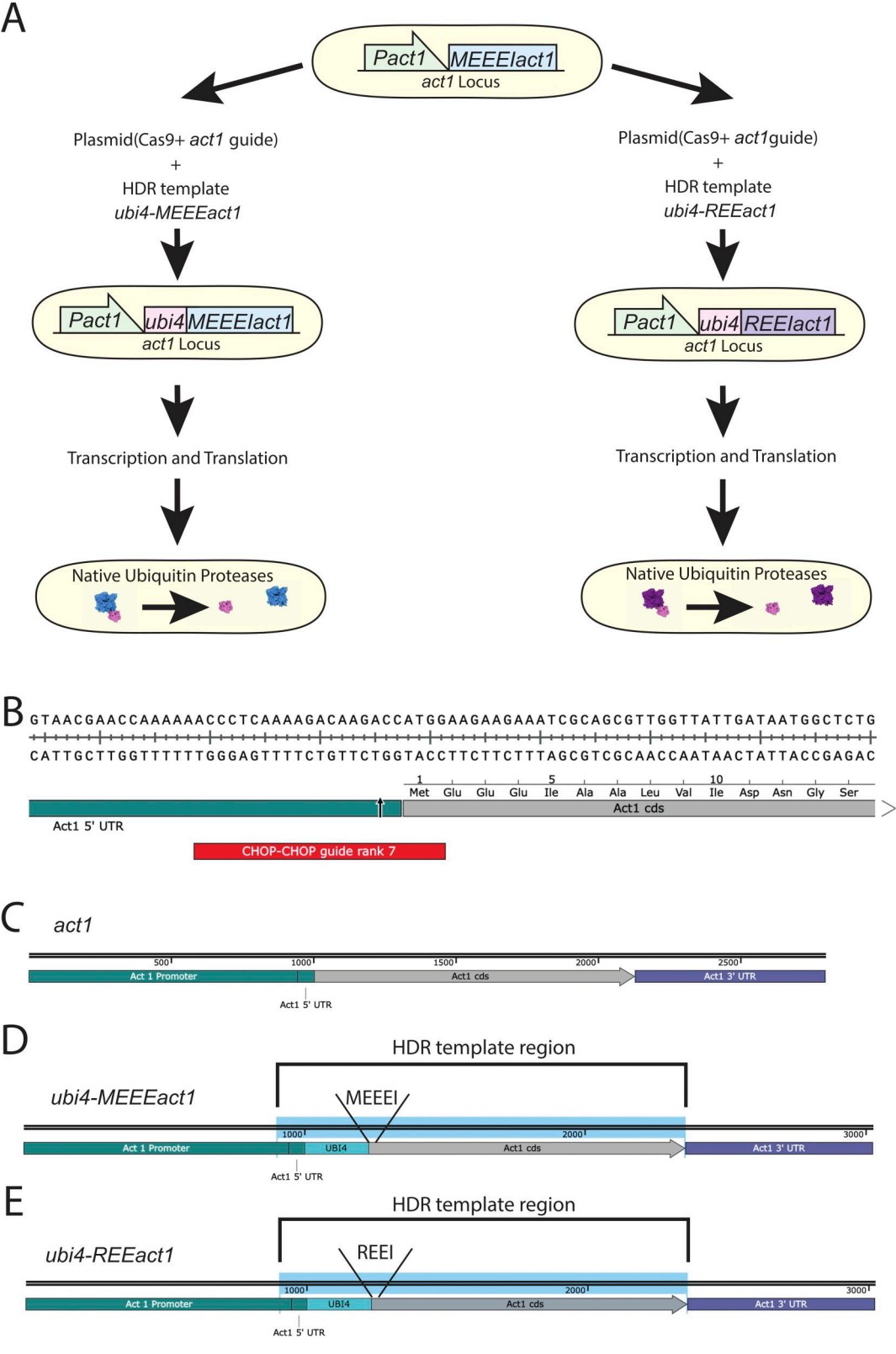

Figure S3. **Approach used to generate the *REEact1* and the control *S. pombe* strains. (A)** Schematic of the approach used to generate the REE and MEEE strains. **(B)** Sequence of the junction between the *act1* promoter and the *act1* coding sequence (cds) showing the location of the chosen guide RNA target (red) designed in CHOP–CHOP and the expected Cas9 cut site (arrow). **(C)** Schematic of the *act1* locus. **(D)** Schematic of the expected *ubi4-MEEEact1* incorporation at the native *act1* locus, including the HDR template region that was used to generate the strains. **(E)** Schematic of the expected *ubi4-REEact1* incorporation at the native *act1* locus, including the HDR template region that was used to generate the strains.

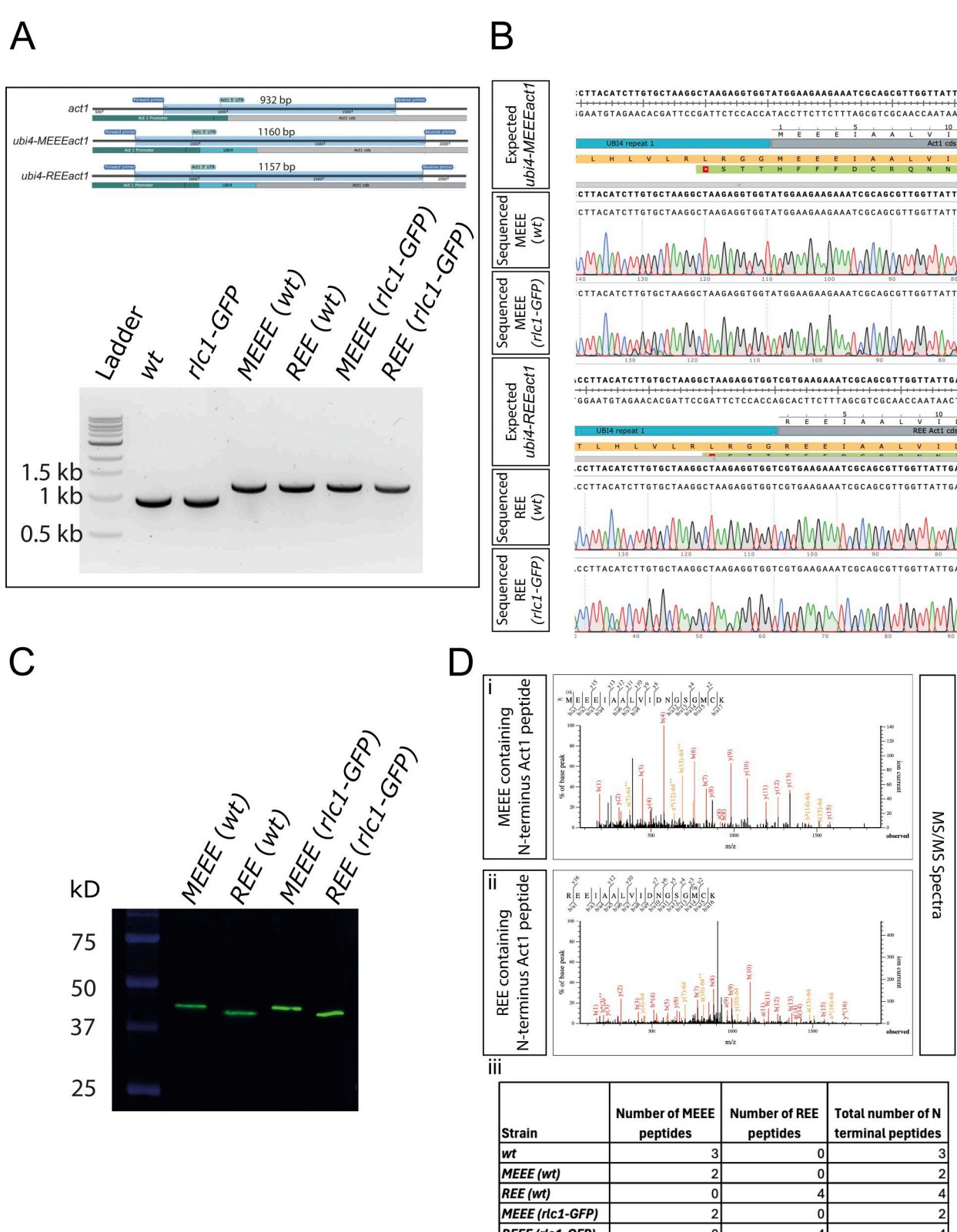

Figure S4. **Validation of the *REEact1* and control *S. pombe* strains. (A)** Schematic of the PCR products and their sizes for the *act1*, *ubi4-MEEEact1*, and *ubi-REEact1* strains using the same pair of primers for genotyping and 1% agarose gel image of the PCR products of the different strains using the genotyping primers. Note the presence of a single band of the expected sizes for all the strains. **(B)** Sequence alignment results for the four *ubi4* strains as compared with their respective expected sequences as derived from SnapGene software. **(C)** Western blot using an anti-actin antibody for the four Ubi4 strains. The actin bands are at the expected size of ~42 kD of the actin product after Ubi4 tag cleavage. Also note the differences in migration of actin between the MEEE and REE strains **(D)** (i) MS/MS spectrum for an MEEE peptide from the MEEE (*wt*) strain. (ii) MS/MS spectrum for an REE peptide from the REE (*wt*) strain. (iii) Table of the number of MEEE or REE peptides observed for the different strains. Source data are available for this figure: SourceData FS4.

Video 1.    **Time-lapse movies of motility assays for the movement of F-actin.** NM myosin II was mixed with phalloidin-labelled filamentous Ac-β-actin and R-β-actin, respectively. Scale bars are 10 µm.

Video 2.    **Time-lapse movies of cells from the MEEE and REE strains in the *rlc1-GFP* background as indicated, showing mCherry-Atb2 (magenta) and Rlc1-3GFP (green).** Images were captured at 3-min intervals. The scale bar is 1 µm. Montages of the individual movies are shown in Fig. 5 A.

**Provided online is Table S1. Table S1 shows cryo-EM data collection, refinement, and validation statistics.**

