## [Peer Review File · The Journal of Cell Biology]

Actin Arginylation Alters Myosin Engagement and F-Actin Patterning despite Structural Conservation

Clyde Pinto, Saskia Bakker, Andrejus Suchenko, Isabella Kolodny, Hamdi Hussain, Tomoyuki Hatano, Karuna Sampath, Krishna Chinthalapudi, Sarah Heissler, Masanori Mishima, and Mohan Balasubramanian

Corresponding Author(s): Mohan Balasubramanian, University of Warwick

Review Timeline:

Submission Date:	2024-09-12
Editorial Decision:	2024-11-13
Revision Received:	2025-08-19
Editorial Decision:	2025-10-01
Revision Received:	2025-10-03

Monitoring Editor: Greg Alushin

Scientific Editor: Dan Simon

Transaction Report:

DOI: <https://doi.org/10.1083/jcb.202409067>

November 13, 2024

Re: JCB manuscript #202409067

Mohan Balasubramanian
University of Warwick

Dear Prof. Balasubramanian,

Thank you for submitting your manuscript entitled "Structure and physiological investigation of arginylated actin." Your manuscript has been assessed by expert reviewers, whose comments are appended below. Thank you for your patience with the peer review process. Although the reviewers express potential interest in this work, significant concerns unfortunately preclude publication of the current version of the manuscript in JCB.

You will see that Reviewers #1&3 question the relevance of mechanistic conclusions that can be drawn from assays employing cardiac myosin II, which has the weakest interactions with the actin N-terminus, and cytoplasmic β -actin, which is different from cardiac actin. This is a key concern which should be addressed with new assays testing the binding and activity of non-muscle myosin II with cytoplasmic β -actin in the presence and absence of arginylation. Reciprocal experiments employing cardiac actin and cardiac myosin II would also have the potential to strengthen the paper. Reviewers #1&2 also request a more thorough assessment of arginylation's effects on actin polymerization and the ATP independence of the R-actin-myosin binding. We also agree that these issues need to be addressed with new data. Reviewer #3 asks for a more detailed description of your R-actin structure and better comparison to published actin structures. This and the requests for more methodological details along with text and figure changes should also be addressed. The Reviewers were split on the value of experiments in the yeast strain that only expresses arginylated actin. We feel that this is a good system for initial analysis of possible gross changes in myosin-based functions. However, the text should include caveats making it clear that this system cannot be used to make definitive conclusions about the role of actin arginylation in cell spreading and lamellipodia formation.

Please let us know if you are able to address the major issues outlined above and wish to submit a revised manuscript to JCB. Note that a substantial amount of additional experimental data likely would be needed to satisfactorily address the concerns of the reviewers. The typical timeframe for revisions is three to four months. If you anticipate any difficulties in meeting this aforementioned revision time limit, please contact us and we can work with you to find an appropriate time frame for resubmission. Please note that papers are generally considered through only one revision cycle, so any revised manuscript will likely be either accepted or rejected.

If you choose to revise and resubmit your manuscript, please also attend to the following editorial points. Please direct any editorial questions to the journal office.

GENERAL GUIDELINES:

Text limits: Character count is < 40,000, not including spaces. Count includes title page, abstract, introduction, results, discussion, and acknowledgments. Count does not include materials and methods, figure legends, references, tables, or supplemental legends.

Figures: Your manuscript may have up to 10 main text figures. To avoid delays in production, figures must be prepared according to the policies outlined in our Instructions to Authors, under Data Presentation, <https://jcb.rupress.org/site/misc/ifora.xhtml>. All figures in accepted manuscripts will be screened prior to publication.

Supplemental information: There are strict limits on the allowable amount of supplemental data. Your manuscript may have up to 5 supplemental figures. Up to 10 supplemental videos or flash animations are allowed. A summary of all supplemental material should appear at the end of the Materials and methods section.

Please note that JCB now requires authors to submit Source Data used to generate figures containing gels and Western blots with all revised manuscripts. This Source Data consists of fully uncropped and unprocessed images for each gel/blot displayed in the main and supplemental figures. Since your paper includes cropped gel and/or blot images, please be sure to provide one Source Data file for each figure that contains gels and/or blots along with your revised manuscript files. File names for Source Data figures should be alphanumeric without any spaces or special characters (i.e., SourceDataF#, where F# refers to the associated main figure number or SourceDataFS# for those associated with Supplementary figures). The lanes of the gels/blots should be labeled as they are in the associated figure, the place where cropping was applied should be marked (with a box),

and molecular weight/size standards should be labeled wherever possible. Source Data files will be made available to reviewers during evaluation of revised manuscripts and, if your paper is eventually published in JCB, the files will be directly linked to specific figures in the published article.

If you choose to resubmit, please include a cover letter addressing the reviewers' comments point by point. Please also highlight all changes in the text of the manuscript.

Regardless of how you choose to proceed, we hope that the comments below will prove constructive as your work progresses. We would be happy to discuss them further once you've had a chance to consider the points raised. You can contact the journal office with any questions at cellbio@rockefeller.edu.

Thank you for thinking of JCB as an appropriate place to publish your work.

Sincerely,

Greg Alushin, PhD
Monitoring Editor
Journal of Cell Biology

Dan Simon, PhD
Scientific Editor
Journal of Cell Biology

Reviewer #1 (Comments to the Authors (Required)):

Understanding how a wide variety of PTMs impact the structure, binding partner interactions and cellular functions of cytoplasmic actins has been a longstanding challenge in the cytoskeleton field. A new method for purifying arginylated actin (R-actin) was used by Pinto et al to determine the high resolution cryoEM structure of arginylated human beta-actin (R-b-actin). Not surprisingly, the R-b-actin filament is found to be highly similar to structure of other actin and it is seen to have a disordered Nter. Wild type b-actin (WT-b) and R-b-actin polymerize equally well while R-b-actin stimulates the actin-activated Mg-ATPase of cardiac myosin about 40% less than WT actin. In the absence of ADP, R-b-actin filaments bind less efficiently to cardiac myosin II bound to coverslip in an in vitro motility chamber. In the presence of ATP, the R-b-actin filaments appear to detach more readily from the myosin and wobble, while the gliding velocity is similar to that of WT actin filaments. These results show that the addition of the positively charged Arg to the Nter of human b-actin modulates the actomyosin interaction. Complementary experiments with *S. pombe* where the single actin gene (*act1*) is engineered to solely express R-actin shows that critical stages during cytokinesis (node coalescence, dwell time, constriction time and rate) proceed more slowly than in cells expressing WT Act1. These cells also have a reduced number of endocytic actin patches compared to WT-actin expressing cells. These results demonstrate the impact of R-actin on the actin-myosin interaction and cytoskeletal dynamics in cells.

The results clearly establish the impact of arginylation of cytoplasmic actins on the actin-myosin II interaction that would be expected to reduce actomyosin contractility. This would likely also be the case for other types of myosins given the generally conserved nature of the actomyosin interface. The in vivo results are consistent with this finding and demonstrate how arginylation can modulate different actin cytoskeletal functions. It is an open question whether or how the modified Nter of cytoplasmic b-actin might impact the interaction with actin binding proteins, but this work sets the stage for future detailed studies of this problem.

The paper initially seems like a bit of a patchwork, mixing structural and biochemical results on human R-b-actin with functional tests of the fission yeast actin cytoskeleton. In the case of the experiments with *S. pombe*, the impact of the modification on several functions is indeed significant but it is not clear how much of the altered function is due to diminished actin-myosin interaction or altered interaction with actin binding proteins. However, the breadth of the work can be seen as a strong point as it shows the impact of R-actin on a critical binding partner (myosin II) in vitro and how R-actin specifically affects several different actin cytoskeletal activities in vivo.

Specific Comments

1) There is a bit of a mismatch in the acto-myosin experiments where the authors investigate the interaction of human cytoplasmic b-actin with cardiac myosin II. While it is true that cardiac actin can be arginylated and actin is quite highly conserved, there are differences between cardiac and cytoplasmic actins and some evidence that the activity of myosins can

vary in assays depending on the actin isoform used. Ideally, one would want to work with the most 'native' system possible, but at a minimum the authors should address the use of cardiac myosin II in place of non-muscle myosin II and how that might, or not, affect the results here.

2) The in vitro assays investigate the effects of R-b-actin on interaction with cardiac myosin II using homogenous actin monomers and filaments. This is certainly quite reasonable for a full understanding of the properties of the R-b-actin but this does not reflect the in vivo state where only a small fraction of R-b-actin is reportedly present in cells (Chen & Kashina, 2019). It would be of interest to see how mixing the R-b-actin with WT actin in native ratios (if known) or different ratios to see how impactful the modified actin could be when mixed with the WT actin.

3) The synthetic lethal experiments with the myo2-E1 allele are interesting and consistent with the observed reduced functionality of R-actin in *S. pombe*. The authors could provide readers with a bit more context for the significance of the change in endocytic patch density in the REE cells. Can this possibly be attributed to a change in the actin-Myo1 interaction or another actin binding protein that is known to maintain normal number of endocytic patches? If so, can this be tested similarly with a synthetic lethal test with a Myo1 allele (if such exists)?

The REE cells in Fig 4C appear to be smaller - has cell size been measured to determine if this might be the case? Is there any change in the size of the patches themselves that might indicate slowing of patch formation?

Minor comments

The authors conclude that there is no significant difference in filament assembly is seen between WT and R-b-actin (Fig 3A). The amount of pelleted actin does look the same in the gel presented but no quantification is provided. The two actins were polymerized using standard conditions, at room temperature for one hour, and then pelleted at high speed. It is possible that polymerization kinetics could be altered and this would be missed in such an assay.

pg 4, bottom right - the authors refer readers to Fig S5B,C,D when describing the mass spec confirmation of the Nter of the MEEE- and REE-actins in the act1 strains but it appears that it should be just Fig S5B,C. Also, they refer readers to Fig S5, E when discussing viability results, but no such panel exists, presumably they meant panel D.

The spot assay for *S. pombe* MEEE (*rlc1-GFP*) and REE (*rlc1-GFP*) strains are said to show that the two do not show any difference in viability or growth, but it looks as if the growth of the REE (*rlc1-GFP*) strain is somewhat compromised as the density of the yeast colonies seems to be smaller.

The authors test for a genetic interaction between the myo2-E1 allele and REE actin and do not find any find that viable colonies with both mutations (Fig S6 - should that be a table?). The authors state that the myo2-E1 mutant is compromised for interaction with wild type actin. The citation provided is Balasubramanian et al 1998 that reports the isolation of the temperature-sensitive allele but not the molecular characterization. This appears to be described in Palani et al (2017).

Supp Video 2 clearly shows the difference in the time for constriction of the contractile ring to be completed for the MEEE (*rlc1-GFP*) and REE (*rlc1-GFP*) strains but it's a bit difficult to know what one is looking at initially as the two movies are right next to each other without any box or separator around the movies to indicate that two separate movies placed side by side are being shown (or so it would seem).

The myosin purification method should state that tissue-purified porcine cardiac myosin was used just to convey to the reader that it is a mix of alpha- and beta-cardiac myosins. The method for making cardiac myosin S1 should be described. Also, if the actin-activated Mg ATPase activity assays were performed using cardiac myosin S1 that should be specified in the text on pg 4 and the legend for Fig 3.

Reviewer #2 (Comments to the Authors (Required)):

Pinto et al. address the question of how arginylation impacts actin structure and function, which is a very important and interesting question. A fair amount of cell biology has been done but biochemistry is sorely lacking. The complement of structural biology, biochemistry, and cell biology leads to a potentially satisfying story. Using yeast as a cell biological model, such that only one type of actin is present for the assays, is a great choice. That said, the analysis of actin dynamics was disappointingly superficial, given the opportunity afforded here and based on the thick cable phenotype observed in yeast.

Major concerns:

pg. 4 The conclusion that arginylation has minimal effect on the in vitro actin polymerization is made based on two very superficial assessments. Using primarily sedimentation, especially at only one concentration of actin, is not appropriate given the tools we have for analysis of actin dynamics. Concluding that the filaments appear similar when bound to myosin ignores the impact of the interaction with myosin, which we later learn is not nothing. More importantly, the fact that these filaments must be

stabilized by phalloidin, which dramatically changes actin dynamics, negates any conclusions. I find no reference to phalloidin stabilization of the filaments but they should not exist at the reported 20-40 nM concentration used in this assay. If filaments were actually stable at this concentration in the absence of phalloidin or some other stabilizer, then something is off about both types of actin. However, we know from this group's previous work that their system generates excellent actin that is indistinguishable from tissue purified act.

Furthermore, because one of the phenotypes observed in yeast is thick cables, measurements of R-actin dynamics with Cdc12 are needed. One could begin to address questions such as, are the cables thicker because of a favorable interaction with formins or because there is more actin available because the R-actin is less well stabilized in patches...?

Pg. 5 Is the actin phalloidin-stabilized in ATPase activities? If not, because we don't know the critical concentration of R-actin, we don't know if this is an appropriate comparison.

Pg. 7 The authors propose that the difference observed in R-actin/myosin interactions is a change in affinity. If this change is independent of ATP, it could be readily tested with cosedimentation assays. The authors report that the number of filaments landing on myosin coated coverslips in the absence of ATP are the same. Shouldn't this be sensitive to a change of affinity? Or is the effect only apparent when nucleotide is present. Please discuss (and possibly test) these matters.

Minor concerns that can be addressed in the writing.

Pg. 3 which residues are included in the comparisons between actin isoforms? Only those resolved in the new structure?

Pg. 3 I struggled with this sentence: "A comparison of the N-terminal residues shows the large number of negatively charged residues in β -actin, which are reduced by the arginylation (see also Fig. 2. C)." A better comparison, without myosin present would be valuable here. Please also tell the reader that the side chains are colored to reflect charge. Finally, how was the structure in 2C made? Was there a template or fitting or? I actually think this figure could focus just on the N-term of actin since the actin-myosin interface is depicted in detail in Fig 3A.

Pg. 4 What should we take home from this: "Despite the higher sequence similarity between the rabbit and chicken actin models, the RMSD between the two at 1.353 Å is lower than between either of those structures and our model of human F-actin."?

Pg. 4 Please be specific about what you are comparing in "Actin arginylation following removal of the N-terminal acetylated aspartate increases the positive electrostatic surface charge by 3 units ." (unmodified, acetylated, or is the result the same? If so, please say so.)

Pg. 4 Fig 3F reports percent of filaments moving, which is not consistent with the following: "Consequently, the number of motile actin filaments was reduced in the R- β -actin sample compared to the Ac- β -actin sample (Fig. 3. F)."

Pg. 7 It's not clear to me whether the N-terminus of actin is more or less resolved in other structures? Please clarify if positions are different or if stability is different?

Reviewer #3 (Comments to the Authors (Required)):

Pinto et al. used cryo-EM to resolve the structure of arginylated ADP-bound F-actin followed by analysis of its actomyosin interactions and influence of yeast actin substitution by the mammalian cytoplasmic arginylated actin on yeast cytoskeleton morphology. I conclude that publication of the current version of the manuscript in JCB is premature for the following reasons:

1. The interpretation and comparison of the R-actin structure is not well detailed. First, I would suggest the authors to provide local resolution map. Next, the authors do not discuss the differences between their R-actin and others appropriately - which regions have higher RMSD? Why rabbit skeletal has enhanced RMSD compared to beta-actin? From the manuscript, the only information that I obtained is that the N-terminus of R-actin and its D-loop are disordered by unknown mechanism(s). How those mechanism(s) may be applicable to the actomyosin interactions except the reduction in the overall charge is left unknown. It would be important for the authors to add more value to their structure.
2. The interaction of the N-terminus of actin with loop 2 of myosin has been reported for many actomyosin complexes. Notably, it was found to differ in those complexes. The authors completely ignore this information in the manuscript and use cardiac myosin isoform which has the weakest interactions with the actin N-terminus. Is it possible that if a correct myosin isoform is used the result may be different?
3. Loop 2 interactions with the actin's N-terminus have been proposed to stabilize the weak bound state of myosin. The authors report that rigor interaction of myosin with actin is not affected by arginylation (ATP free conditions). Meanwhile, in presence of ATP (weak and strong bound myosins) they report frequent detachments of R-actin filaments from myosin. Hence, the authors' data points to the importance of actin's N-terminus interaction with the myosin loop2 during weak binding state. This should be

discussed.

5. The authors decided to use yeast system to elucidate the effects of arginylation. Yeast cytoskeleton is very different from the mammalian cells, hence, the effects that the authors report in yeast may be not applicable to the mammalian system where the role of arginylation was proposed for cell spreading and lamellipodia formation. The authors should discuss it.

Reviewer #1 (Comments to the Authors (Required)):

Understanding how a wide variety of PTMs impact the structure, binding partner interactions and cellular functions of cytoplasmic actins has been a longstanding challenge in the cytoskeleton field. A new method for purifying arginylated actin (R-actin) was used by Pinto et al to determine the high resolution cryoEM structure of arginylated human beta-actin (R-b-actin). Not surprisingly, the R-b-actin filament is found to be highly similar to structure of other actin and it is seen to have a disordered Nter. Wild type b-actin (WT-b) and R-b-actin polymerize equally well while R-b-actin stimulates the actin-activated Mg-ATPase of cardiac myosin about 40% less than WT actin. In the absence of ADP, R-b-actin filaments bind less efficiently to cardiac myosin II bound to coverslip in an in vitro motility chamber. In the presence of ATP, the R-b-actin filaments appear to detach more readily from the myosin and wobble, while the gliding velocity is similar to that of WT actin filaments. These results show that the addition of the positively charged Arg to the Nter of human b-actin modulates the actomyosin interaction. Complementary experiments with *S. pombe* where the single actin gene (*act1*) is engineered to solely express R-actin shows that critical stages during cytokinesis (node coalescence, dwell time, constriction time and rate) proceed more slowly than in cells expressing WT Act1. These cells also have a reduced number of endocytic actin patches compared to WT-actin expressing cells. These results demonstrate the impact of R-actin on the actin-myosin interaction and cytoskeletal dynamics in cells.

The results clearly establish the impact of arginylation of cytoplasmic actins on the actin-myosin II interaction that would be expected to reduce actomyosin contractility. This would likely also be the case for other types of myosins given the generally conserved nature of the actomyosin interface. The in vivo results are consistent with this finding and demonstrate how arginylation can modulate different actin cytoskeletal functions. It is an open question whether or how the modified Nter of cytoplasmic b-actin might impact the interaction with actin binding proteins, but this work sets the stage for future detailed studies of this problem.

The paper initially seems like a bit of a patchwork, mixing structural and biochemical results on human R-b-actin with functional tests of the fission yeast actin cytoskeleton. In the case of the experiments with *S. pombe*, the impact of the modification on several functions is indeed significant but it is not clear how much of the altered function is due to diminished actin-myosin interaction or altered interaction with actin binding proteins. However, the breadth of the work can be seen as a strong point as it shows the impact of R-actin on a critical binding partner (myosin II) in vitro and how R-actin specifically affects several different actin cytoskeletal activities in vivo.

We thank the referee for their detailed and constructive comments on this manuscript. We have addressed all the points raised through the inclusion of several new experiments, notable of which is the full characterization of R-actin-myosin-II interactions with non-muscle myosin-II (while moving the cardiac myosin motility assay experiments in the original submission to the supplemental material). Genetic and physiological experiments suggested have also been carried out.

Specific Comments

1) There is a bit of a mismatch in the acto-myosin experiments where the authors investigate the interaction of human cytoplasmic β -actin with cardiac myosin II. While it is true that cardiac actin can be arginylated and actin is quite highly conserved, there are differences between cardiac and cytoplasmic actins and some evidence that the activity of myosins can vary in assays depending on the actin isoform used. Ideally, one would want to work with the most 'native' system possible, but at a minimum the authors should address the use of cardiac myosin II in place of non-muscle myosin II and how that might, or not, affect the results here.

RESPONSE: We thank the reviewer for this comment. We have now performed *in vitro* experiments with a phosphorylated active version of non-muscle myosin II. These *in vitro* experiments show reduced stimulation of the myosin ATPase activity and increased filament detachment during myosin-driven surface gliding when R-actin filaments are used, consistent with our previous experiments using cardiac myosin. We have included these results in Figures 3 and S2. We have now described the non-muscle myosin II experiments in the manuscript and supplement, while the cardiac myosin findings have been described as a table in Fig. S2 E.

2) The *in vitro* assays investigate the effects of R- β -actin on interaction with cardiac myosin II using homogenous actin monomers and filaments. This is certainly quite reasonable for a full understanding of the properties of the R- β -actin but this does not reflect the *in vivo* state where only a small fraction of R- β -actin is reportedly present in cells (Chen & Kashina, 2019). It would be of interest to see how mixing the R- β -actin with WT actin in native ratios (if known) or different ratios to see how impactful the modified actin could be when mixed with the WT actin.

RESPONSE: We have tested a 1:1 mixture of acetylated and arginylated actin in the *in vitro* motility assay. No obvious impact of partial arginylation on actin motility has been detected. However, this might be due to the limitation in the sensitivity of our method. Inside cells, the concentration of actin modifications can be location dependent and very high at a given location such as at the leading edge of migrating cells. In the Discussion section of the revised manuscript, we mention this point: "Interestingly, when both actin isoforms were mixed 1:1 no difference in filament landing to the surface and difference in filament motility was detected compared to Ac- β -actin samples only (data not shown). However, R- β -actin concentration in different parts of cells may vary. Filaments containing mostly R- β -actin might be more common on the leading edge of the cell facilitating migration of the cell (Pavlyk et al., 2018)." (Page 9 lines 263-267)

3) The synthetic lethal experiments with the myo2-E1 allele are interesting and consistent with the observed reduced functionality of R-actin in *S. pombe*. The authors could provide readers with a bit more context for the significance of the change in endocytic patch density in the REE cells. Can this possibly be attributed to a change in the actin-Myo1 interaction or another actin binding protein that is known to maintain normal number of endocytic patches? If so, can this be tested similarly with a synthetic lethal test with a Myo1 allele (if such exists)? We have now performed a more detailed quantitation of actin patches in the two strains. We have included this along with the context of actin patch dependence on Arp2/3 complex.

RESPONSE: We have carried out a series of experiments based on this suggestion. In short, the strong deleterious interaction we observed between *REEact1* and *myo2-E1* is specific for this combination and is not observed in the *REEact1 myo1Δ* strain. Although minor additive effects cannot be ruled out, synthetic lethality is not observed in this combination. We also did not find any synthetically lethal interactions between *REEact1* and formins (*cdc12-112* and *for3Δ*), *wsp1Δ*, and *arp3-c1*. (Page 7 lines 224-226)

The REE cells in Fig 4C appear to be smaller - has cell size been measured to determine if this might be the case? Is there any change in the size of the patches themselves that might indicate slowing of patch formation?

RESPONSE: We have now measured both cell size and patch size and number. We find that the cell length, volume and surface area are reduced by 15.2%, 10.1% and 11.4% respectively and cell width is increased by 3.3%. This information is provided in Figure 4 C. Furthermore, we found that the number of actin patches per cell was reduced by 19.5% and we also found a reduction in patch area of 10.2%, which is also described in Figure 4 E.

Minor comments

The authors conclude that there is no significant difference in filament assembly is seen between WT and R-b-actin (Fig 3A). The amount of pelleted actin does look the same in the gel presented but no quantification is provided. The two actins were polymerized using standard conditions, at room temperature for one hour, and then pelleted at high speed. It is possible that polymerization kinetics could be altered and this would be missed in such an assay.

RESPONSE: Polymerisation of these actins has been reported previously in Chin et al., 2022. They find that the rate of arginylated actin polymerisation is lower than acetylated actin, although it reaches a similar plateau level within the time frame relevant to our assays (~1 hour). The sedimentation of the filaments in Fig S1A (now performed with 4 different concentrations) and the length distribution in Fig. S2 C confirm this and guarantee that any influence detected in our *in vitro* assays is not primarily due to the difference in actin polymerization. We clarified this in the main text (lines 131-137 on Pg5).

pg 4, bottom right - the authors refer readers to Fig S5B,C,D when describing the mass spec confirmation of the Nter of the MEEE- and REE-actins in the act1 strains but it appears that it should be just Fig S5B,C. Also, they refer readers to Fig S5, E when discussing viability results, but no such panel exists, presumably they meant panel D.

RESPONSE: We thank the reviewer for pointing these errors out. We have now completely changed the organisation of the figures and reassigned the panel numbers correctly.

The spot assay for *S. pombe* MEEE (*rlc1-GFP*) and REE (*rlc1-GFP*) strains are said to show that the two do not show any difference in viability or growth, but it looks as if the growth of

the REE (*rlc1-GFP*) strain is somewhat compromised as the density of the yeast colonies seems to be smaller.

RESPONSE: We have now performed a more detailed analysis of the growth of the wild type strains. This shows that there is a slight difference between the REE and MEEE strains in terms of growth. This becomes pronounced when *rlc1-3GFP* is introduced into this background (Fig. 4B). Previous work has shown that *rlc1-3GFP* mimics a weak allele of *rlc1* and this is consistent with the deleterious interactions between *REEact1* and *myo2-E1*.

The authors test for a genetic interaction between the *myo2-E1* allele and REE actin and do not find any find that viable colonies with both mutations (Fig S6 - should that be a table?). The authors state that the *myo2-E1* mutant is compromised for interaction with wild type actin. The citation provided is Balasubramanian et al 1998 that reports the isolation of the temperature-sensitive allele but not the molecular characterization. This appears to be described in Palani et al (2017).

RESPONSE: We thank the reviewer for pointing out this mistake. The correct citation has now been added to the text. (Page 7 line 218-219)

Supp Video 2 clearly shows the difference in the time for constriction of the contractile ring to be completed for the MEEE (*rlc1-GFP*) and REE (*rlc1-GFP*) strains but it's a bit difficult to know what one is looking at initially as the two movies are right next to each other without any box or separator around the movies to indicate that two separate movies placed side by side are being shown (or so it would seem).

RESPONSE: A white line separator is now placed between the two movies.

The myosin purification method should state that tissue-purified porcine cardiac myosin was used just to convey to the reader that it is a mix of alpha- and beta-cardiac myosins. The method for making cardiac myosin S1 should be described.

RESPONSE: We have now changed this in the methods (Pg. 15, line 495). Cardiac myosin S1 in the experiments in the original manuscript was commercial, but this data is now removed.

Also, if the actin-activated Mg ATPase activity assays were performed using cardiac myosin S1 that should be specified in the text on pg 4 and the legend for Fig 3.

RESPONSE: New non-muscle myosin data have now been added and both the figures and text have been updated to specify which myosin was used in each experiment. This new data can be found in Figure 3 and S2.

Reviewer #2 (Comments to the Authors (Required)):

Pinto et al. address the question of how arginylation impacts actin structure and function, which is a very important and interesting question. A fair amount of cell biology has been done but biochemistry is sorely lacking. The complement of structural biology, biochemistry, and

cell biology leads to a potentially satisfying story. Using yeast as a cell biological model, such that only one type of actin is present for the assays, is a great choice. That said, the analysis of actin dynamics was disappointingly superficial, given the opportunity afforded here and based on the thick cable phenotype observed in yeast.

We thank the referee for the very useful feedback. We appreciate the fact that the referee likes the yeast work. Several new experiments have been performed, characterizing better the genetic interactions and the phenotypes described. The relevant quantifications have been provided in the revised manuscript. Furthermore, we have carried out a series of *in vitro* experiments using non-muscle myosin-II as indicated in the first point in the main concerns.

Major concerns:

pg. 4 The conclusion that arginylation has minimal effect on the *in vitro* actin polymerization is made based on two very superficial assessments. Using primarily sedimentation, especially at only one concentration of actin, is not appropriate given the tools we have for analysis of actin dynamics. Concluding that the filaments appear similar when bound to myosin ignores the impact of the interaction with myosin, which we later learn is not nothing.

RESPONSE: Polymerization properties of arginylated actin have been previously characterized in detail by Chin et al. (2022), who reported that while arginylated actin polymerizes more slowly than acetylated actin, it reaches a similar plateau within the time frame relevant to our assays (~1 hour). The sedimentation of the filaments in Fig S1A (now performed with 4 different concentrations) and the length distribution in Fig. S2 C confirm this and guarantee that any influence detected in our *in vitro* assays is not primarily due to the difference in actin polymerization. We clarified this in the main text: "These observations are consistent with the results of the bulk sedimentation assay (Fig. S1A) and saturation levels of polymerisation reported previously (Chin et al., 2022), though the kinetics of polymerisation was reduced by a factor of ~1.5-1.8. These actin preparations with comparable polymerisation levels and filament length distributions were used in the following *in vitro* assays" (page 5, lines 131-137)

More importantly, the fact that these filaments must be stabilized by phalloidin, which dramatically changes actin dynamics, negates any conclusions. I find no reference to phalloidin stabilization of the filaments but they should not exist at the reported 20-40 nM concentration used in this assay. If filaments were actually stable at this concentration in the absence of phalloidin or some other stabilizer, then something is off about both types of actin. However, we know from this group's previous work that their system generates excellent actin that is indistinguishable from tissue purified act.

RESPONSE: Although the *in vitro* motility assay in the original manuscript was performed without phalloidin, no significant depolymerization was observed during the first 15 minutes following dilution. This is consistent with a previous report showing that aged actin filaments depolymerize only very slowly (Kueh et al., 2008). As the reviewer suggested, to minimize any potential influence of post-translational modifications on filament stability, we included phalloidin in the revised *in vitro* motility experiments with non-muscle myosin II (Fig. 3).

Furthermore, because one of the phenotypes observed in yeast is thick cables, measurements of R-actin dynamics with Cdc12 are needed. One could begin to address questions such as, are the cables thicker because of a favorable interaction with formins or because there is more actin available because the R-actin is less well stabilized in patches...?

RESPONSE: We have now additionally performed crosses to test for synthetic lethality between the *REEact1* strain and the formins *for3Δ* and *cdc12-112*. The strains we used were not synthetic lethal with the REE *act1* at the permissive temperature. However, we found a reduction in the number and size of actin patches in the *REEact1* strain. We believe that as described by us and Kovar laboratory, reduced incorporation of R-actin into actin patches increases the pool available for incorporation into actin cables (Balasubramanian et al., 1996; Kadzik et al., 2020).

Pg. 5 Is the actin phalloidin-stabilized in ATPase activities? If not, because we don't know the critical concentration of R-actin, we don't know if this is an appropriate comparison.

RESPONSE: We have now performed the new experiments with non-muscle myosin with phalloidin stabilized actin filaments. The previous studies with cardiac myosin were not phalloidin stabilized. The new results are provided in Figure 3 and Figure S2.

Pg. 7 The authors propose that the difference observed in R-actin/myosin interactions is a change in affinity. If this change is independent of ATP, it could be readily tested with cosedimentation assays. The authors report that the number of filaments landing on myosin coated coverslips in the absence of ATP are the same. Shouldn't this be sensitive to a change of affinity? Or is the effect only apparent when nucleotide is present. Please discuss (and possibly test) these matters.

RESPONSE: In the ATPase cycle, the myosin motor domain interacts strongly with actin when it is nucleotide-free or bound to ADP, and weakly when bound to ATP or ADP-Pi. In the absence of ATP, the strong binding through the main actin-binding surface likely dominates, making this interaction relatively insensitive to modifications at the actin N-terminus. By contrast, in the presence of ATP, myosin II spends most of its cycle in the weakly bound state (as it is a low-duty-ratio motor), which reduces the overall affinity via the main binding surface. Under these conditions, additional interactions — such as electrostatic interactions between the positively charged loop of myosin and the negatively charged actin N-terminal tail — may become more important.

We speculate that this explains why no difference is observed in filament landing in the absence of ATP, while actin N-terminal modifications affect gliding stability and myosin–actin interactions only when ATP is present. This is now discussed in page 9 lines 258-267 of the revised manuscript.

Minor concerns that can be addressed in the writing.

Pg. 3 which residues are included in the comparisons between actin isoforms? Only those resolved in the new structure?

RESPONSE: We have now included this in the manuscript from line 115 on Pg.4. "We used CCP4 Superpose by Secondary Structure Matching (Krissinel and Henrick, 2004) to establish matching residue ranges. Residues 3-374 in the structure presented here were matched with residues 4-375 in structures 6DJO and 3J8I."

Pg. 3 I struggled with this sentence: "A comparison of the N-terminal residues shows the large number of negatively charged residues in β -actin, which are reduced by the arginylation (see also Fig. 2. C)."

RESPONSE: Thanks for pointing this out. We have now clarified this sentence (see line 110-113 Pg. 4.).

A better comparison, without myosin present would be valuable here. Please also tell the reader that the side chains are colored to reflect charge. Finally, how was the structure in 2C made? Was there a template or fitting or?

RESPONSE: We have now removed this figure from the manuscript. The interaction of actin and myosin is described in Figure 3. The reference to the colouring of charges is therefore not relevant.

Pg. 4 What should we take home from this: "Despite the higher sequence similarity between the rabbit and chicken actin models, the RMSD between the two at 1.353 Å is lower than between either of those structures and our model of human F-actin."?

RESPONSE: There was a typo in the sentence and this could have caused part of the confusion. Inadvertently, we used lower rather than higher, which has now been rectified (line 121, Pg.4).

The RMSD value represented in a pairwise manner are chicken-Rabbit (1.353Å), chicken-R-actin (0.965Å) and rabbit-R-actin (1.248Å).

These are statements of facts that we discovered and are reporting. More details are provided in our response to referee 3 (point #1).

Pg. 4 Please be specific about what you are comparing in "Actin arginylation following removal of the N-terminal acetylated aspartate increases the positive electrostatic surface charge by 3 units ." (unmodified, acetylated, or is the result the same? If so, please say so.)

RESPONSE: We are comparing AcDDD and RDD (without acetylation). We have now changed the sentence to "Arginylation of β -actin at the N-terminus exposed by removal of the acetylated aspartate increases the electrostatic surface charge by 3 as compared to Ac- β -actin" (lines 124-125 on Pg. 5)

Pg. 4 Fig 3F reports percent of filaments moving, which is not consistent with the following: "Consequently, the number of motile actin filaments was reduced in the R- β -actin sample compared to the Ac- β -actin sample (Fig. 3. F)."

RESPONSE: We thank the reviewer for pointing out that the previous figure 3. F showed percentage and not number of motile actin filaments. We have now performed experiments with NMIIA. We have described the relevant experiments as follows “While slightly more R- β -actin filaments were observed per field of view (Fig. 3 C), we observed a lower percentage of motile R-actin filaments compared to Ac- β -actin filaments (Fig. 3 D).” (lines 148-150, Pg 5)

Pg. 7 It's not clear to me whether the N-terminus of actin is more or less resolved in other structures? Please clarify if positions are different or if stability is different?

RESPONSE: The N-terminus of actin is typically unresolved in most cryo-EM structures due to its inherent flexibility. However, in our previous work, we successfully resolved the N-terminus for several actin isoforms, providing new structural insights (Arora et al., 2023). Notably, the positioning of the actin N-terminus appears to be dynamic and is predicted to vary depending on its interaction partners. For instance, when actin is bound to myosin, the N-terminus can adopt distinct conformations influenced by the charge distribution of myosin's loop-2 residues (von der Ecken et al., 2016). This structural plasticity may play a critical role in modulating actin's interactions and functions within different cellular contexts.

Reviewer #3 (Comments to the Authors (Required)):

Pinto et al. used cryo-EM to resolve the structure of arginylated ADP-bound F-actin followed by analysis of its actomyosin interactions and influence of yeast actin substitution by the mammalian cytoplasmic arginylated actin on yeast cytoskeleton morphology. I conclude that publication of the current version of the manuscript in JCB is premature for the following reasons:

Many thanks to the referee for their critique and suggestions. We have addressed these in full below.

1. The interpretation and comparison of the R-actin structure is not well detailed. First, I would suggest the authors to provide local resolution map. Next, the authors do not discuss the differences between their R-actin and others appropriately - which regions have higher RMSD? Why rabbit skeletal has enhanced RMSD compared to beta-actin? From the manuscript, the only information that I obtained is that the N-terminus of R-actin and its D-loop are disordered by unknown mechanism(s). How those mechanism(s) may be applicable to the actomyosin interactions except the reduction in the overall charge is left unknown. It would be important for the authors to add more value to their structure.

a. which regions have higher RMSD?

RESPONSE:

There are multiple elements to this comment #1, which we address in full below.

The local resolution map has been provided in Figure S1B.

We have now included the statement on Page 4 lines 108-110 "As expected, higher RMSD is seen in the D-loop region around residue 50, and two solvent-exposed loops around residues

146 and 245, corresponding to the areas with lower resolution as shown in the local resolution map".

We assume the referee is asking the RMSD of rabbit actin vs R-beta-actin, the focus of this work. We believe much of the RMSD differences can be attributed to differences in the D-loop and the loops mentioned above. We believe that we need to have structures of R-actin in different nucleotide states (as in (Galkin et al., 2015) and (Chou and Pollard, 2019)) to discern the basis of disordering in this and other regions of actin filaments, although one can imagine that hydrolysis of ATP may be a major contributor.

The charge of the R-actin N-terminus affecting myosin interaction is consistent with the multiple F-actin-myosin structural complexes. But this had never been directly tested before. Even though our structure doesn't resolve the N-terminus, our *in vitro* experiments using purified R-actin and non-muscle myosin-II and *in vivo* experiments with fission yeast mutants fill this gap for the first time and present a full picture of how actin arginylation impacts actomyosin interactions. These elements are expanded in #3 below.

The modifications carried out in rewriting this revised version, we believe largely improves the "value-addition" to our structure.

2. The interaction of the N-terminus of actin with loop 2 of myosin has been reported for many actomyosin complexes. Notably, it was found to differ in those complexes. The authors completely ignore this information in the manuscript and use cardiac myosin isoform which has the weakest interactions with the actin N-terminus. Is it possible that if a correct myosin isoform is used the result may be different?

RESPONSE: We repeated the *in vitro* experiments with non-muscle myosin-II and observed very similar effects of arginylation in both the ATPase and motility assays, i.e., reduced stimulation of myosin ATPase and increased filament detachment/wobbling in the motility assay, despite the different levels of ATPase and gliding velocity. Considering the biological relevance of the arginylation in non-muscle cells, we now show these new data with non-muscle myosin-II in the main Fig. 3 and the motility assay data with cardiac myosin are in Fig. S2

3. Loop 2 interactions with the actin's N-terminus have been proposed to stabilize the weak bound state of myosin. The authors report that rigor interaction of myosin with actin is not affected by arginylation (ATP free conditions). Meanwhile, in presence of ATP (weak and strong bound myosins) they report frequent detachments of R-actin filaments from myosin. Hence, the authors' data points to the importance of actin's N-terminus interaction with the myosin loop2 during weak binding state. This should be discussed.

RESPONSE: We appreciate this suggestion and now discussed this point in the Discussion of the revised manuscript. "In general, myosin binds actin strongly when nucleotide-free or ADP-bound, and weakly when bound to ATP or ADP-Pi. As a low-duty-ratio motor, myosin-II spends most of its ATPase cycle in the weak-binding state. We speculate that the electrostatic interaction through the N-terminal tail of actin plays a role in loosely anchoring myosin heads in the weak or none-binding state, while its impact is minimal for the myosin in the strong-binding state (such as in the absence of ATP)". (page 8,9 and lines 258-263)

5. The authors decided to use yeast system to elucidate the effects of arginylation. Yeast cytoskeleton is very different from the mammalian cells, hence, the effects that the authors report in yeast may be not applicable to the mammalian system where the role of arginylation was proposed for cell spreading and lamellipodia formation. The authors should discuss it.

RESPONSE: The point the referee raises is interesting. In balance, we believe that though not physiological, the *S. pombe* strain we have developed provides a “synthetic” live cell system to investigate if there are overt effects in cells solely reliant on R-actin for viability. We have discussed the pros and cons of the use of fission yeast in our discussion (Pages 9,10; lines 289-296).

References

- Arora, A.S., H.L. Huang, R. Singh, Y. Narui, A. Suchenko, T. Hatano, S.M. Heissler, M.K. Balasubramanian, and K. Chinthalapudi. 2023. Structural insights into actin isoforms. *Elife*. 12.
- Balasubramanian, M.K., A. Feoktistova, D. McCollum, and K.L. Gould. 1996. Fission yeast Sop2p: a novel and evolutionarily conserved protein that interacts with Arp3p and modulates profilin function. *The EMBO Journal*. 15:6426-6437.
- Chin, S.M., T. Hatano, L. Sivashanmugam, A. Suchenko, A.S. Kashina, M.K. Balasubramanian, and S. Jansen. 2022. N-terminal acetylation and arginylation of actin determines the architecture and assembly rate of linear and branched actin networks. *J Biol Chem*. 298:102518.
- Chou, S.Z., and T.D. Pollard. 2019. Mechanism of actin polymerization revealed by cryo-EM structures of actin filaments with three different bound nucleotides. *Proc Natl Acad Sci U S A*. 116:4265-4274.
- Galkin, V.E., A. Orlova, M.R. Vos, G.F. Schroder, and E.H. Egelman. 2015. Near-atomic resolution for one state of F-actin. *Structure*. 23:173-182.
- Kadzik, R.S., K.E. Homa, and D.R. Kovar. 2020. F-Actin Cytoskeleton Network Self-Organization Through Competition and Cooperation. *Annu Rev Cell Dev Biol*. 36:35-60.
- Krissinel, E., and K. Henrick. 2004. Secondary-structure matching (SSM), a new tool for fast protein structure alignment in three dimensions. *Acta Crystallogr D Biol Crystallogr*. 60:2256-2268.
- Kueh, H.Y., W.M. Brieher, and T.J. Mitchison. 2008. Dynamic stabilization of actin filaments. *Proc Natl Acad Sci U S A*. 105:16531-16536.
- Pavlyk, I., N.A. Leu, P. Vedula, S. Kurosaka, and A. Kashina. 2018. Rapid and dynamic arginylation of the leading edge beta-actin is required for cell migration. *Traffic*. 19:263-272.
- von der Ecken, J., S.M. Heissler, S. Pathan-Chhatbar, D.J. Manstein, and S. Raunser. 2016. Cryo-EM structure of a human cytoplasmic actomyosin complex at near-atomic resolution. *Nature*. 534:724-728.

October 1, 2025

RE: JCB Manuscript #202409067R

Mohan Balasubramanian
University of Warwick

Dear Prof. Balasubramanian,

Thank you for submitting your revised manuscript entitled "Actin Arginylation Alters Myosin Engagement and F-Actin Patterning despite Structural Conservation." We would be happy to publish your paper in JCB pending the minor text and figure changes recommended by the reviewers and final revisions necessary to meet our formatting guidelines (see details below).

A. MANUSCRIPT ORGANIZATION AND FORMATTING:

1) Text limits: Character count for Reports is < 20,000, not including spaces. Count includes title page, abstract, introduction, results & discussion, and acknowledgments. Count does not include materials and methods, figure legends, references, tables, or supplemental legends.

****Reports must have a single 'Results and Discussion' section.****

2) Figure formatting: Reports may have up to 5 main text figures. Scale bars must be present on all microscopy images, including inset magnifications. Molecular weight or nucleic acid size markers must be included on all gel electrophoresis. Also, please avoid pairing red and green for images and graphs to ensure legibility for color-blind readers. If red and green are paired for images, please ensure that the particular red and green hues used in micrographs are distinctive with any of the colorblind types. If not, please modify colors accordingly or provide separate images of the individual channels.

3) Statistical analysis: Error bars on graphic representations of numerical data must be clearly described in the figure legend. The number of independent data points (n) represented in a graph must be indicated in the legend. Please indicate whether 'n' refers to technical or biological replicates (i.e. number of analyzed cells, samples or animals, number of independent experiments). If independent experiments with multiple biological replicates have been performed, we recommend using distribution-reproducibility SuperPlots (please see Lord et al., JCB 2020) to better display the distribution of the entire dataset, and report statistics (such as means, error bars, and P values) that address the reproducibility of the findings.

Statistical methods should be explained in full in the materials and methods. For figures presenting pooled data the statistical measure should be defined in the figure legends. Please also be sure to indicate the statistical tests used in each of your experiments (both in the figure legend itself and in a separate methods section) as well as the parameters of the test (for example, if you ran a t-test, please indicate if it was one- or two-sided, etc.). Also, if you used parametric tests, please indicate if the data distribution was tested for normality (and if so, how). If not, you must state something to the effect that "Data distribution was assumed to be normal but this was not formally tested."

4) Materials and methods: Should be comprehensive and not simply reference a previous publication for details on how an experiment was performed. Please provide full descriptions (at least in brief) in the text for readers who may not have access to referenced manuscripts. The text should not refer to methods "...as previously described." Please also indicate the type of membrane used for immunoblotting as well as describe acquisition and quantification methods. Centrifugation speeds should be given in rcf not rpm values.

5) For all cell lines, vectors, strains, constructs/cDNAs, etc. - all genetic material: please include database / vendor ID (e.g. Addgene, ATCC, etc.) or if unavailable, please briefly describe their basic genetic features, even if described in other published work or gifted to you by other investigators (and provide references where appropriate). Please be sure to provide the sequences for all of your oligos: primers, si/shRNA, RNAi, gRNAs, etc. in the materials and methods. You must also indicate in the methods the source, species, and catalog numbers/vendor identifiers (where appropriate) for all of your antibodies, including secondary. If antibodies are not commercial, please add a reference citation if possible.

6) Microscope image acquisition: The following information must be provided about the acquisition and processing of images:

- a. Make and model of microscope
- b. Type, magnification, and numerical aperture of the objective lenses

- c. Temperature
- d. Imaging medium
- e. Fluorochromes
- f. Camera make and model
- g. Acquisition software
- h. Any software used for image processing subsequent to data acquisition. Please include details and types of operations involved (e.g., type of deconvolution, 3D reconstitutions, surface or volume rendering, gamma adjustments, etc.).

7) References: There is no limit to the number of references cited in a manuscript. References should be cited parenthetically in the text by author and year of publication. Abbreviate the names of journals according to PubMed.

8) Supplemental materials: Reports may have up to 5 supplemental figures and 10 videos. Please also note that tables, like figures, should be provided as individual, editable files. A summary of all supplemental material should appear at the end of the Materials and methods section. Please include one brief sentence per item.

9) Video legends: Should describe what is being shown, the cell type or tissue being viewed (including relevant cell treatments, concentration and duration, or transfection), the imaging method (e.g., time-lapse epifluorescence microscopy), what each color represents, how often frames were collected, the frames/second display rate, and the number of any figure that has related video stills or images.

10) eTOC summary: A ~40-50 word summary that describes the context and significance of the findings for a general readership should be included on the title page. The statement should be written in the present tense and refer to the work in the third person. It should begin with "First author name(s) et al..." to match our preferred style.

11) Conflict of interest statement: JCB requires inclusion of a statement in the acknowledgements regarding competing financial interests. If no competing financial interests exist, please include the following statement: "The authors declare no competing financial interests." If competing interests are declared, please follow your statement of these competing interests with the following statement: "The authors declare no further competing financial interests."

12) A separate author contribution section is required following the Acknowledgments in all research manuscripts. All authors should be mentioned and designated by their first and middle initials and full surnames. We encourage use of the CRediT nomenclature (<https://casrai.org/credit/>).

13) ORCID IDs: ORCID IDs are unique identifiers allowing researchers to create a record of their various scholarly contributions in a single place. Please note that ORCID IDs are required for all authors. At resubmission of your final files, please be sure to provide your ORCID ID and those of all co-authors.

14) JCB requires authors to submit Source Data used to generate figures containing gels and Western blots with all revised manuscripts. This Source Data consists of fully uncropped and unprocessed images for each gel/blot displayed in the main and supplemental figures. For assays performed using capillary electrophoresis and/or immunoassay-based detection, authors should instead provide the electropherogram graph(s) for each experiment, plotting fluorescence/chemiluminescence intensity vs. molecular weight/size. Since your paper includes cropped gel and/or blot images, please be sure to provide one Source Data file for each figure gels, blots, and/or capillary electrophoresis assays along with your revised manuscript files. File names for Source Data figures should be alphanumeric without any spaces or special characters (i.e., SourceDataF#, where F# refers to the associated main figure number or SourceDataFS# for those associated with Supplementary figures). For traditional gels and blots, the lanes of the gels/blots should be labeled as they are in the associated figure, the place where cropping was applied should be marked (with a box), and molecular weight/size standards should be labeled wherever possible. For capillary electrophoresis assays, each trace in the graph should be color-coded and labeled to indicate which protein, gene, or sample is being measured (please try to avoid red/green combinations to accommodate our color-blind readers).

Source Data files will be directly linked to specific figures in the published article. Source Data Figures should be provided as individual PDF files (one file per figure). Authors should endeavor to retain a minimum resolution of 300 dpi or pixels per inch. Please review our instructions for export from Photoshop, Illustrator, and PowerPoint here: <https://rupress.org/jcb/pages/submission-guidelines#revised>

15) Journal of Cell Biology now requires a data availability statement for all research article submissions. These statements will be published in the article directly above the Acknowledgments. The statement should address all data underlying the research presented in the manuscript. Please visit the JCB instructions for authors for guidelines and examples of statements at (<https://rupress.org/jcb/pages/editorial-policies#data-availability-statement>).

B. FINAL FILES:

Thank you for your attention to these final processing requirements. Please revise and format the manuscript and upload materials within 7 days. If you need an extension for whatever reason, please let us know and we can work with you to determine a suitable revision period.

Thank you for this interesting contribution, we look forward to publishing your paper in Journal of Cell Biology.

Sincerely,

Greg Alushin, PhD
Monitoring Editor
Journal of Cell Biology

Dan Simon, PhD
Scientific Editor
Journal of Cell Biology

Reviewer #1 (Comments to the Authors (Required)):

The authors have nicely addressed this reviewer's comments. Overall, the work is quite interesting and highlights how small changes at the Nter of actin can impact the actomyosin interaction and, as a result, critical cellular functions such as cytokinesis and endocytosis.

Very minor comments -

Please specify that the NM II isoform used in the assays presented here is NM IIA.

line 217 The authors discuss their results with an *S. pombe* myosin II mutant, myo2-E1, without describing that for readers. While the mutant designation may readily indicate to some that it's a myosin II mutant, those who work outside of *S. pombe* might appreciate having this mutant briefly described.

lines 258, 259 The authors state that 'myosin binds actin strongly when nucleotide-free or ADP-bound.... Please provide a citation in support of this statement. Additionally, is this true for all myosins or, more importantly, for the myosin II and myosin I myosins that are relevant for this work?

Reviewer #2 (Comments to the Authors (Required)):

The paper is ready for publication.

Minor:

For the non-yeast division aficionados, more description of the CAR parameters would be helpful. Also, marking key events in figure 5A would help the reader.

Reviewer #3 (Comments to the Authors (Required)):

I commend the authors for their thoughtful revision. Congratulations on a nice piece of work.